# Genome-wide association mapping of total antioxidant capacity, phenols, tannins, and flavonoids in a panel of *Sorghum bicolor* and *S. bicolor* × *S. halepense* populations using multi-locus models

**Ephrem Habyarimana**[1]*, **Michela Dall'Agata**[1], **Paolo De Franceschi**[1], **Faheem S. Baloch**[2]

**1** CREA Research Center for Cereal and Industrial Crops, Bologna, Italy, **2** Department of Field Crops, Faculty of Agricultural and Natural Sciences, Abant Izzet Baysal University, Bolu, Turkey

* ephrem.habyarimana@crea.gov.it

## Abstract

Sorghum is widely used for producing food, feed, and biofuel, and it is increasingly grown to produce grains rich in health-promoting antioxidants. The conventional use of grain color as a proxy to indirectly select against or for antioxidants polyphenols in sorghum grain was hampered by the lack of consistency between grain color and the expected antioxidants concentration. Marker-assisted selection built upon significant loci identified through linkage disequilibrium studies showed interesting potential in several plant breeding and animal husbandry programs, and can be used in sorghum breeding for consumer-tailored antioxidant production. The purpose of this work was therefore to conduct genome-wide association study of sorghum grain antioxidants using single nucleotide polymorphisms in a novel diversity panel of *Sorghum bicolor* landraces and *S. bicolor* × *S. halepense* recombinant inbred lines. The recombinant inbred lines outperformed landraces for antioxidant production and contributed novel polymorphism. Antioxidant traits were highly correlated and showed very high broad-sense heritability. The genome-wide association analysis uncovered 96 associations 55 of which were major quantitative trait loci (QTLs) explaining 15 to 31% of the observed antioxidants variability. Eight major QTLs localized in novel chromosomal regions. Twenty-four pleiotropic major effect markers and two novel functional markers (Chr9_1550093, Chr10_50169631) were discovered. A novel pleiotropic major effect marker (Chr1_61095994) explained the highest proportion ($R^2 = 27$–31%) of the variance observed in most traits evaluated in this work, and was in linkage disequilibrium with a hotspot of 19 putative glutathione S-transferase genes conjugating anthocyanins into vacuoles. On chromosome four, a hotspot region was observed involving major effect markers linked with putative MYB-bHLH-WD40 complex genes involved in the biosynthesis of the polyphenol class of flavonoids. The findings in this work are expected to help the scientific community particularly involved in marker assisted breeding for the development of sorghum cultivars with consumer-tailored antioxidants concentration.

**Data Availability Statement:** All relevant data are within the paper and its Supporting Information files.

**Funding:** Part of this work was supported (beneficiary: first author) by the European Union project Data-driven Bioeconomy (www.databio.eu), GA number: 732064 (H2020-ICT-2016-1 - innovation action), and the project Risorse GeneticheVegetali (RGV/FAO) 2014e2016 of the Ministero delle Politiche Agricole, Alimentari, Forestali e del Turismo, Rome, Italy; Beneficiary: E. H. The funders had no role in study design, data collection and analysis, decision to publish, or preparation of the manuscript.

**Competing interests:** The authors have declared that no competing interests exist.

# Introduction

Sorghum (*Sorghum bicolor* (L.) Moench) is one of the world's most important crops grown for food, feed, and biofuel [1]. Sorghum was traditionally a staple food for hundreds of millions of people in Africa and Asia, but it is becoming more popular worldwide, including in developed countries, for its uses in food industry to satisfy a rise in demand for specialty grains, especially those that are gluten-free and rich in health-promoting compounds [2]. Moreover, the importance of sorghum in agricultural ecosystems is expected to keep rising because this crop presents high resilience to the predicted food security threatening scenarios related to climate change such as drought and heat [3], it is adapted under lower and higher latitudes [4], and it outperforms several crop species in terms of productivity and energy balance [1].

Like in other cultivated species, sorghum genetic improvement for production and quality traits including antioxidants concentration in grains, can benefit from controlled hybridizations with wild relatives [5]. Breeding efforts are being increasingly committed worldwide to introgress perenniality from wild species into domesticated sorghum with the aim of increasing crop sustainability as well as food security [6,7]. *Sorghum halepense* (L.) Pers. [6,8–10] and *S. propinquum* (Kunth) Hitchc [11,12] are frequently used to transfer perenniality into *Sorghum bicolor*, but *S. halepense* confers a more robust perenniality [13], and hence it was used as wild donor of this trait in this study. *Sorghum halepense*, commonly called 'Johnsongrass' is a natural allotetraploid (2n = 40) most likely originated by the spontaneous hybridization between *S. bicolor* (2n = 20) and *S. propinquum* (2n = 20) followed by chromosome doubling [14]. It can be hybridized with either induced tetraploids or cytoplasmic-genetic male sterile diploids of cultivated sorghum, originating in both cases mainly tetraploid progenies [9,15], although diploid descendants can also occur [16,17].

In 2014 we initiated a breeding program aimed at developing perennial grain sorghum for use in specialty foods. Under the human nutrition perspectives, sorghum grain shows important health-promoting properties and it is a rich source of antioxidants such as polyphenols, carotenoids, as well as micro- and macronutrients [18,19] all of which constitute key ingredients in several human healthy diets. Indeed, sorghum grain exhibits the highest values of total antioxidant capacity among several cereals including wheat, rice, oats, barley and maize [2,20].

The antioxidant properties of sorghum grain are mostly dependent on the contents of phenols (polyphenols) which in turn are classified in phenolic acids (benzoic or cinnamic acid derivatives) and flavonoids. These compounds in sorghum grains are found in the seed pericarp (external coat) and testa (the layer between pericarp and endosperm). Polyphenols are a large class of phytonutrients that are beneficial to human health [21–23]. They have the potential to protect against several chronic diseases including cancer, type 2 diabetes, heart disease, and other living cell damaging oxidative reactions produced by free radicals, through antioxidant reactions [24]. In sorghum grain, the flavonoids are mostly represented by condensed tannins (proanthocyanidins) and anthocyanins [2,20]. The condensed tannins (hereafter referred to as tannins) are high-molecular-weight polymers of catechins and epicatechins produced through the flavonoid pathway and are the only type of tannins that have been found in sorghum [25]. In sorghum seeds, they play a role in seed dormancy [26], protect against grain mold [27] and bird and insect predation [28]. In addition, due to tannin's ability to bind proteins and minerals, tannin-containing sorghum is reported to decrease feed efficiency in some animal species, and this property can help fight obesity and several other degenerative afflictions in humans [29]. It is worth noticing that tannins are found in grains, such as sorghum with a pigmented testa layer, some finger millets, and barley, but not in major cereal crops, such as rice, wheat, and maize [20]. Tannin content in sorghum grains is generally much higher than in other cultivated fruits, nuts, and grains, and both tannin and non-tannin types

occur in nature and in cultivated sorghums, but wild sorghums are mostly tannin types [20]. Given that one of the objectives of breeding has long been to select against the tannins in the caryopsis [30], particularly because of its astringency and its negative effect in animal feeding, the persistent residual tannin trait in cultivated sorghum needs to be explained. As suggested by Wu et al. [30], it is believed that natural selection retained a certain tannin content in domesticated sorghum as these compounds conferred sorghum resistance to frequent grain molds and bird damages. As the human interest shifts towards maintaining or increasing these healthy compounds in grain, it can be expected that fortifying sorghum grain with increased contents in condensed tannins is going to be one of the major objectives in several sorghum breeding programs.

Conventionally, grain color was used to indirectly select against or for polyphenols in sorghum [31]. A comprehensive account on sorghum grain color and the genetics and physiology of phenols can be found in Rhodes et al. [31]. According to these authors [31] and as described below, a number of classical loci reported to govern grain color and testa, govern also the presence or absence of polyphenols in sorghum. For instance, genotypes with dominant alleles at the B loci (B1- and B2-) show pigmented testa and proanthocyanidins in the testa layer. Dominant alleles at the spreader (S) and at the B1 and B2 loci, produce proanthocyanidins in the pericarp and in the testa layer, and can result in a brown-colored grain. The pericarp base color is red, yellow, or white, and is controlled by the R and Y loci. Interestingly, the spreader, intensifier (I) and mesocarp thickness (Z) loci can modify the base pericarp color, leading to colors ranging from brilliant white to black with several shades of red, yellow, pink, orange and brown [31]. Endosperm color in sorghum shows a range of color from white to yellow; recent studies showed that yellow endosperm in sorghum is correlated with carotenoids, which are mainly lutein, zeaxantin and β-carotene [32]. It can be inferred that these colored phenotypes confounded conventional breeders in their effort to control polyphenols in sorghum grains, and might have contributed, in addition to the natural selection described above [30], to the persistence of polyphenols in grains of cultivated sorghum crop.

Marker-assisted selection (MAS) built upon significant loci identified through linkage and, particularly, linkage disequilibrium (genome-wide association studies, GWAS) studies showed interesting potential in several plant breeding and animal husbandry programs, and can be used in sorghum breeding programs for consumer-tailored antioxidant production [31]. The existing linkage and linkage disequilibrium studies aimed at dissecting the genetic basis of antioxidants in sorghum have relied on cultivated *S. bicolor* populations and resulted in several quantitative trait loci (QTLs) most of which with small effects on antioxidants levels [31–38], which makes MAS less effective for these compounds. Specifically, QTLs for grain color, antioxidant capacity, polyphenols, tannins, anthocyanin, proanthocyanins and 3-deoxyantocyanidins contents have been reported and are recorded in the Sorghum QTL Atlas platform (http://aussorgm.org.au/sorghum-qtl-atlas/) [39]. While the genes underpinning most QTL effects on antioxidant activity have yet to be clearly identified, two major genes, *Yellow seed1* and *Tannin1* were proved to be involved in the biosynthesis of polyphenols (Fig 1) in sorghum grain [30,40]. They encode for a MYB (*Yellow seed1*) and a WD40 (*Tannin1*) transcription factors, the latter having homology to *Arabidopsis transparent testa glabra 1* (TTG1). The genomic positions of *Yellow seed1* and *Tannin1* (on chromosomes 1 and 4, respectively) and their allelic polymorphisms were confirmed to be correlated with tannins concentration in sorghum grains [30,40]. Many other genes have been associated to tannin variation and grain antioxidant activity, including two homologs of *Arabidopsis thaliana transparent testa* (TT) genes TT10 and TT4 [31,37].

This work aimed at conducting genome-wide study of sorghum grain antioxidants using single nucleotide polymorphism (SNP) markers in a novel diversity panel consisting of a

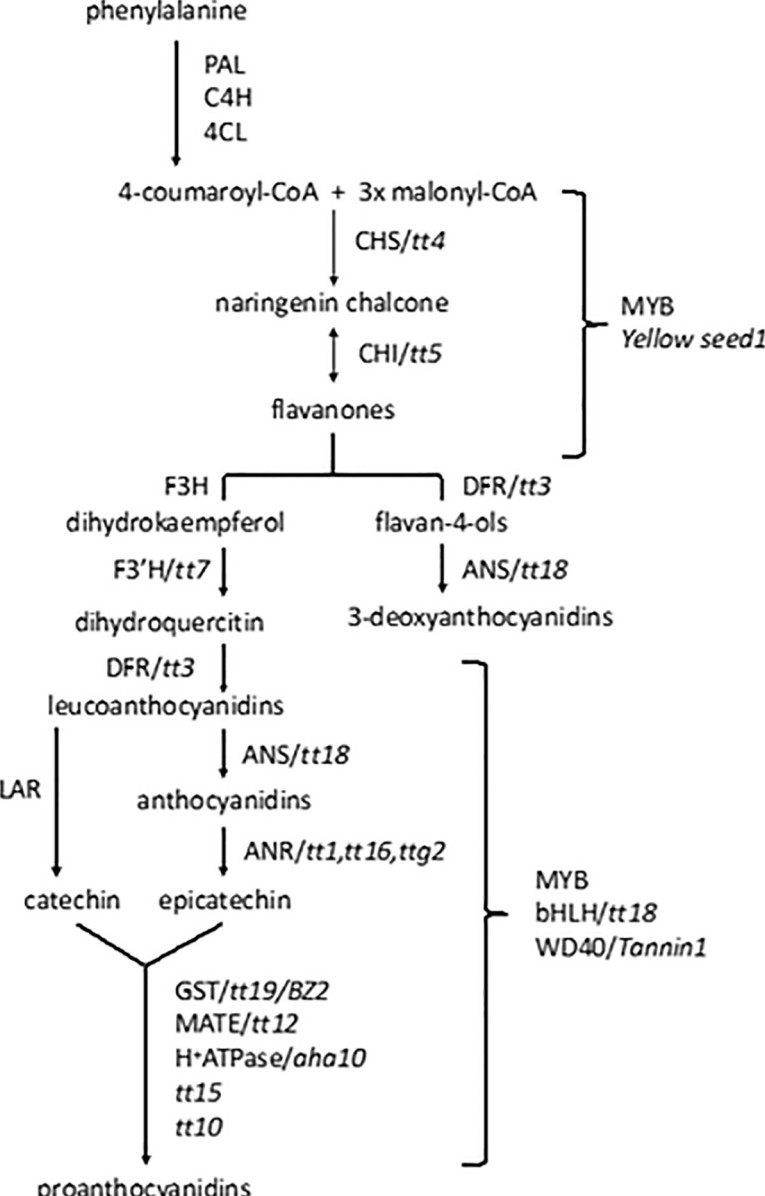

**Fig 1. Biosynthesis and regulation of flavonoids.** PAL = phenylalanine ammonia lypase, C4H = trans-cinnamate 4-monoxygenase, 4CL = 4-coumarate CoA ligase; CHS = chalcone synthase; CHI = chalcone isomerase; F3H = chalcone-flavanone isomerase hydroxylase; F3'H = flavanone 3'-hydroxylase; DFR = dihydroflavonol-4-reductase; ANS = anthocyanidin synthase; ANR = anthocyanidin reductase; GST = glutathione-S-transferase; MATE = multidrug and toxic efflux transporter; LAR = leucoanthocyanidin reductase; TT = transparent testa; TTG = transparent testa glabra.

combination of *Sorghum bicolor* landraces and a progeny derived from *S. bicolor* × *S. halepense* (SBxSH) hybridizations. To the best of our knowledge, it is the first time such a panel is used in association analyses. In virtue of intrinsic genetic properties of the plant materials used in this work, it was hypothesized that the landraces and, particularly, the *S. halepense* genome would constitute a good source of antioxidants and novel useful marker loci for the association studies in sorghum grain, all of which can help discover causal genetic factors underpinning antioxidant activity. In SBxSH combinations, the wild genome of *S. halepense* represents an

untapped reservoir and a good source of genetic diversity that can be used in conventional and molecular breeding. On the other hand, the *S. bicolor* genotypes were derived from African and Asian landraces, and are expected to harbor a high level of genetic diversity for breeding purposes inasmuch as Africa and Asia represent, respectively, the primary and secondary sorghum centers of diversity [8].

## Materials and methods

### Plant materials

Sorghum materials used in this work consisted of 114 genotypes, of which 95 and 19 were, respectively, *Sorghum bicolor* (SB) and a mixture of recombinant inbred lines derived from *S. bicolor* × *S. halepense* (SBxSH) controlled hybridizations at different levels of filial progeny. These populations were described in previous works [1,8]. *Sorghum biclor* genotypes were annual sorghums selections derived from landraces from Africa and Asia. The SBxSH lines were selections from annual/perennial (A/P) crosses and A/P backcrosses to annual recurrent parents (A*2/P; BC1), perennial/perennial (P/P) and annual/perennial//perennial (A/P//P) crosses. The annual parents were induced tetraploids (2n = 40), standard diploid (2n = 20), genetic male-sterile, and cytoplasmic-genetic male-sterile inbred *S. bicolor* lines. Perennial parents consisted of a *S. halepense* plant and tetraploid lines obtained by hybridizing induced tetraploid sorghum plants with *S. halepense*. Open-field trials for these sorghum populations were run in 2014 and 2015 in the CREA-CI's (CREA Research Center for Cereal and Industrial Crops) experimental station of Anzola (Bologna, Italy), using augmented randomized complete block design with 6 controls (checks) and 6 blocks [41]. Seeds were harvested at physiological maturity as indicated by the appearance of the black layer (hilum) in the caryopses at the base of the panicle.

### Total phenolic, tannins, flavonoids and antioxidant capacity determination

Total phenols, tannins, flavonoids and antioxidant capacity (TAC) were analytically determined as described in our previous works [2,42]. Briefly, a 10 g sample from each genotype was ground using a 0.5 mmm sieve Cyclotec Udy Mill (UDY Corporation), moisture determined in an oven overnight at 105˚C, and antioxidants and TAC analyzed in duplicate using 100 mg of each sample. For the phenolic compounds the absorbance of samples was measured at 750 nm and expressed as gallic acid equivalents (g GAE $kg^{-1}$ dry mass basis). For condensed tannins and total flavonoids assays, the absorbances were measured at 500 nm and 510 nm, respectively, and expressed as μg CE (catechin equivalents) $g^{-1}$ dry mass basis. Total antioxidant content was determined using the 2,20-azino-bis/3-ethylbenzthiazo- line-6-sulphonic acid (ABTS) assay and expressed as mmol TE (Trolox equivalents) $kg^{-1}$ dry mass basis.

### DNA extraction

Sorghum seeds (5–20 per sample) were sown in peat, watered, and were treated with a fungicide (Ortiva, Syngenta, 1ml/L) and an insecticide (Confidor, Bayer, 0.75 ml/L) to protect young plantlets from pathogens and pests. For part of samples grown in winter, seeds were treated with a seed-coating fungicide (Celest, Syngenta, 4 ml/L in water) and allowed to germinate on wet filter paper within petri dishes laid in the incubator Venticell 111 (MMM group) at 25˚C for 4–6 days. 1–3 healthy plantlets (nearly 10–30 cm tall) or 3–5 germinated seeds were collected for each sample and DNA was extracted using the GeneJET Plant Genomic DNA Purification Kit (ThermoFisher Scientific), following manufacturer's instructions. DNA concentration and purity were evaluated by a Tecan Infinite M200Pro spectrophotometer

(Tecan Group Ltd., Switzerland), while DNA integrity was checked through 1% agarose gel electrophoresis containing 10 µl/L GelRed (Biotium) as fluorescent dye. For each DNA sample, an aliquot of 60 µl at a concentration ≥ 10 ng/µl was used for downstream analyses.

## Whole-genome genotyping-by-sequencing SNP genotyping

The panel of 114 sorghum lines evaluated in this work for antioxidants production was genotyped using a genotyping-by-sequencing (GBS) strategy. The sorghum panel was split in two groups as part of larger sets of 184 and 196 individuals that were genotyped separately as batch 1 and batch 2. The methylation sensitive restriction enzyme ApeKI was used for library preparation, and GBS was carried out on an Illumina HiSeq X Ten platform by BGI Hong Kong Company Limited. The sequencing reads were aligned to the sorghum reference genome (Sorghum_bicolor NCBIv3) to enable variants discovery. The two batches yielded two respective matrices of 933,020 and 919,485 markers, and were delivered as separate VCF files which were subsequently merged into a single matrix using VCFtools (https://vcftools.github.io/index. html) [43] resulting in a total of 1,252,091 loci. Marker quality control criteria were then applied to the merged dataset considering only samples having phenotypic and marker data. The filters were implemented in VCF tools to restrict the dataset to high quality standards including biallelic SNPs only, minor allele frequency (MAF) ≥ 0.05, site quality or the Phred-scaled probability that reference/alternative alleles polymorphism exists at a given site given the sequencing data Q≥40 (i.e., ≥99.99% base call accuracy), and missing genotypes (NA) ≤ 20%. The final working matrix consisting of 61,976 high-quality SNPs was used in this work for genome-wide association analyses.

## Genome-wide association assessment

GWAS was conducted using the statistical genetics package Genome Association and Prediction Integrated Tool (GAPIT) [44]. Since population and family genetic structures can interfere with GWAS bringing about spurious candidate marker loci [45,46], we aimed at detecting and visualizing the existing genetic structure using the principal component analysis and the genomic relationship matrix [47]. The genomic relationship matrix and the top three principal components that explained most of the observed phenotypic variance in the trait of interest, were used to control for population and family structure. Two multi-locus GWAS algorithms, FarmCPU [48] and SUPER [49], were used to identify significant QTLs for four sorghum grain antioxidant properties. The Fixed and random model Circulating Probability Unification (FarmCPU) method is currently a commonly used approach, that improves statistical power compared to the existing GWAS methods.

FarmCPU effectively eliminates confounding between test markers and kinship, controls false positives as well as false negatives, dividing Multiple Loci Linear Mixed Model (MLMM) into a Fixed Effect Model (FEM) and a Random Effect Model (REM) and using them iteratively [48]. The settlement of mixed linear models under progressively exclusive relationship (SUPER) is a powerful method that was developed to solve the computational issues of the mixed linear models (MLM), and the preconditional requirements (number of SNPs be less than the number of individuals in order to derive a rank-reduced relationship) of the FaST linear mixed models (FaST-LMM) method [49]. The GWAS model procedures were implemented by solving the below linear mixed model equation [49]:

$$y = X\beta + Zu + \varepsilon$$

where $y$ is a vector of observed phenotypes; $\beta$ is an unknown vector containing fixed effects, including the genetic markers, population structure (Q), and the intercept; $u$ is an unknown

vector of random additive genetic effects from multiple background QTLs in the individuals; $X$ and $Z$ are the known design matrices; $\varepsilon$ is the unobserved vector of residuals. The population structure (Q) is accounted for using principal component. The $u$ and $\varepsilon$ vectors are assumed to be normally distributed with a null mean and a variance of

$$var \begin{bmatrix} u \\ \varepsilon \end{bmatrix} = \begin{bmatrix} G & 0 \\ 0 & R \end{bmatrix}$$

with $G = \sigma_g^2 K$, where $\sigma_g^2$ is the additive genetic variance, and $K$ the kinship through which the information about the relationships among individuals is conveyed. $K$ is used as the variance-covariance matrix between the individuals. Homogeneous variance is assumed for the residual effects namely, $R = \sigma_e^2 I$, where $\sigma_e^2$ is the residual variance and $I$ the identity matrix and unknown residual variance This mixed model equation was adapted to perform GWAS using the full model, accounting for kinship as well as population structure confounding effects.

The significantly associated QTLs were determined by the P-value less than 0.01/m, m being the number of markers [50]; no multiple test correction was required for the multi-locus methods implemented in this work because all markers were fitted to a single model and all effects were estimated and tested simultaneously. Fitness of the GWAS models for all traits was evaluated using Quantile-Quantile (Q-Q) plots of the observed vs. expected–log10(p) values which should follow a uniform distribution under the null hypothesis [46]. To characterize novel QTLs (novel markers) and QTLs overlapping with those previously identified, the position of significant markers was compared to the confidence interval of known QTLs for polyphenol-related traits (3-deoxyanthocyanidins, anthocyanin level, proanthocyanidins, polyphenol content, tannin content, antioxidant activity, grain color) retrieved from the Sorghum QTL atlas [39].

To get functional insights into candidate markers, we searched the vicinity of the GWAS significant markers for antioxidant-relevant 'a priori' and 'a posteriori' candidate genes using gene annotation and ontology information in Phytozome (https://phytozome.jgi.doe.gov/) [51]. The extent of the region flanking (upstream and downstream from the SNP position) significant markers and within which candidate genes were identified, was determined by analyzing the genome-wide LD decay which was 500 Kb in this work at the cut-off $r^2 = 0.1$. All genes involved in polyphenols metabolic pathway were considered 'a priori' candidates, including those genes that were reported in previous works [31–38]. Genes showing homology to the Peroxidases, Laccase and Catechol oxidases families were also included in a priori candidates as they are involved in the oxidation of flavonoids [52]. Regardless of their annotation and putative function, all genes containing a significant marker within their sequence were considered 'a posteriori' candidates.

## Statistical analysis

Statistical inferences for the antioxidant traits and the diagrams produced in this work were performed using appropriate routines called from the R, a statistical computing language and environment [53]. The broad sense heritability was estimated as the ratio of genotypic variance to total phenotypic variance, using variance components estimated by fitting appropriate linear mixed effects model to the data and considering genotypes and replications as random effects [54] as briefly axplained below. Variance components and trait broad-sense heritability (repeatability, hereafter referred to as heritability) were estimated by fitting the linear mixed model equation $y_{ij} = \mu + g_i + e_{ij}$ for $i = 1,\ldots,s$ genotypes, $j = 1,\ldots,n_i$ replicates (year) for genotype $i$, $y_{ij}$ is the response variable for genotype $i$ in replicate (year) $j$; it was assumed $g_i \sim N(0, \sigma_g^2)$

and $e_i \sim N(0, \sigma_e^2)$. Yearly adjusted means were used and the model was fitted with restricted maximum likelihood using the R package lme4. The heritability was derived through the formula $\sigma_g^2/(\sigma_g^2 + \sigma_e^2/n_r)$ where $\sigma_g^2$, $\sigma_e^2$ and $n_r$ are the genetic (genotypic) variance, residual variance, and the number of replications (years), respectively. The genomic relationship matrix relationship matrix was computed as suggested by VanRaden [47]

# Results

## Phenotypic variability and heritability

The descriptive statistics including post hoc analytic results of the contents of the antioxidants measured in the sorghum populations evaluated in this work, are depicted graphically using quartiles (Fig 2). The statistical inferences showed that SBxSH lines outperformed SB populations in terms of total antioxidant capacity (TAC, 59.16 vs. 40.17 mmol TE kg$^{-1}$ dm$^{-1}$), and the contents of tannins (7,358.57 vs. 3,053.25 $\mu$g CE g$^{-1}$ dm$^{-1}$), phenols (6.56 vs. 3.93 g GAE kg$^{-1}$ dm$^{-1}$), and flavonoids (6,738.39 vs. 3,869.66 $\mu$g CE g$^{-1}$ dm$^{-1}$). A wide range of the measurements was observed and reflected the existence of a good phenotypic variability. In the entire sorghum panel, the range of the total antioxidant capacity (TAC), phenols (FEN), tannins (TAN), and flavonoids (FLA) was, respectively, 6.89–172.02 mmol TE kg$^{-1}$ dm$^{-1}$, 0.6–20.73 g GAE kg$^{-1}$ dm$^{-1}$, 0–27,138.56 $\mu$g CE g$^{-1}$ dm$^{-1}$, and 0–22,606.18 $\mu$g CE g$^{-1}$ dm$^{-1}$. In SB, this range was 6.89–147.41 mmol TE kg$^{-1}$ dm$^{-1}$, 0.6–15.44 g GAE kg$^{-1}$ dm$^{-1}$, 0–17,147.57 $\mu$g CE g$^{-1}$ dm$^{-1}$, 0–18,795.57 $\mu$g CE g$^{-1}$ dm$^{-1}$, while in SBxSH the range was 13.58–172.02 mmol TE kg$^{-1}$ dm$^{-1}$, 1.45–20.73 g GAE kg$^{-1}$ dm$^{-1}$, 0–27,138.56 $\mu$g CE g$^{-1}$ dm$^{-1}$, and 1,122.59–22,606.18 $\mu$g CE g$^{-1}$ dm$^{-1}$. The broad-sense heritability was very high with values greater than 0.99 (Table 1) in all the traits. The pairwise Pearson correlation coefficients among the four antioxidant traits (Fig 3H) were positive and very high (r>0.90) and ranged from 0.92 (TAN-TAC) to 0.98 (TAC-FEN).

## Population structure, genotypic and allelic properties

The population structure was analyzed using principal component and genomic relationship (hereafter referred to as G matrix) approaches. The two methods produced similar results, and therefore we used the genomic relationship matrix to better visualize the details of the structure. The heatmap of the G matrix (Fig 4) distinguished between SB and SBxSH subpopulations. In the SB subpopulation, two major groups were observed made up of 36 (just below SBxSH) and 61 (bottom) individuals. The diagonal elements of the G matrix were equal to 1 among SB subpopulation, while these elements were > 1.5 in the SBxSH subpopulation, underscoring that SBxSH was definitely a different population with higher rate of heterozygotes relative to the *S. bicolor* cluster [55]. The frequency distribution of the marker genotypes used in this work are summarized in the histogram in Fig 5. The pattern of the heterozygosity distribution was favorable for conducting GWAS as the homozygotes for the reference allele were the most frequent followed by the homozygotes for the alternative allele, while the heterozygotes were rare. The heterozygosity rate of an individual is the proportion of heterozygous (carrying two different alleles of a specific SNP) genotypes. In GWAS, high levels of heterozygosity within an individual might be an indication of low sample quality whereas low levels of heterozygosity may be due to inbreeding [56]. The distribution of the reference and alternative alleles, and the polymorphism information content (PIC) were presented in Fig 6. The reference allele was negatively skewed in the entire panel and in the SB and SBxSH subpopulations, with the mean smaller than the median, while the alternative allele and the PIC were skewed right, with the mean greater than the median. The statistical dispersion as indicated by the

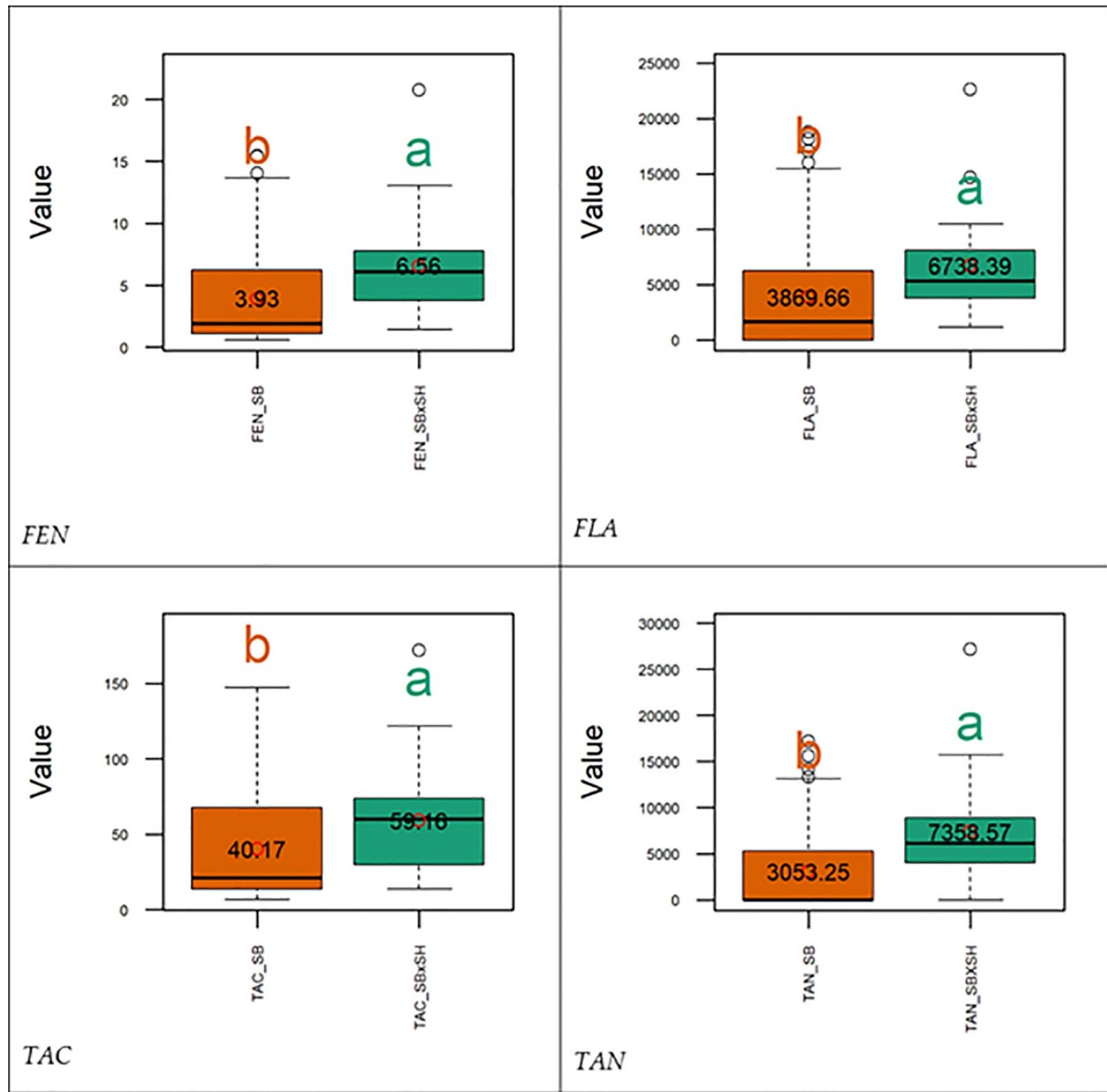

**Fig 2. Distribution and comparison of antioxidants contents in *Sorghum bicolor* and *Sorghum bicolor* × *Sorghum halepense* recombinant inbred lines.** Open dot and the numbers inside the rectangles (interquartile ranges) represent, respectively, the position of mean and the mean of the trait of intertest. The horizontal bar inside the rectangles represents the median of the trait of interest. Within the same trait, means with different letters are significantly different at the 5% level using the Tukey's HSD (honestly significant difference) test. Refer to text for the description of the traits.

**Table 1. Broad-sense heritability of the antioxidants measured in the sorghum panel.**

| Trait | Environmental variance | Genotypic variance | Broad-sense Heritability |
|---|---|---|---|
| Total antioxidant capacity | 0.0046 | 1.0071 | 0.9978 |
| Phenols | 0.0026 | 1.0091 | 0.9987 |
| Tannins | 0.0027 | 1.009 | 0.9987 |
| Flavonoids | 0.0063 | 1.0052 | 0.9969 |

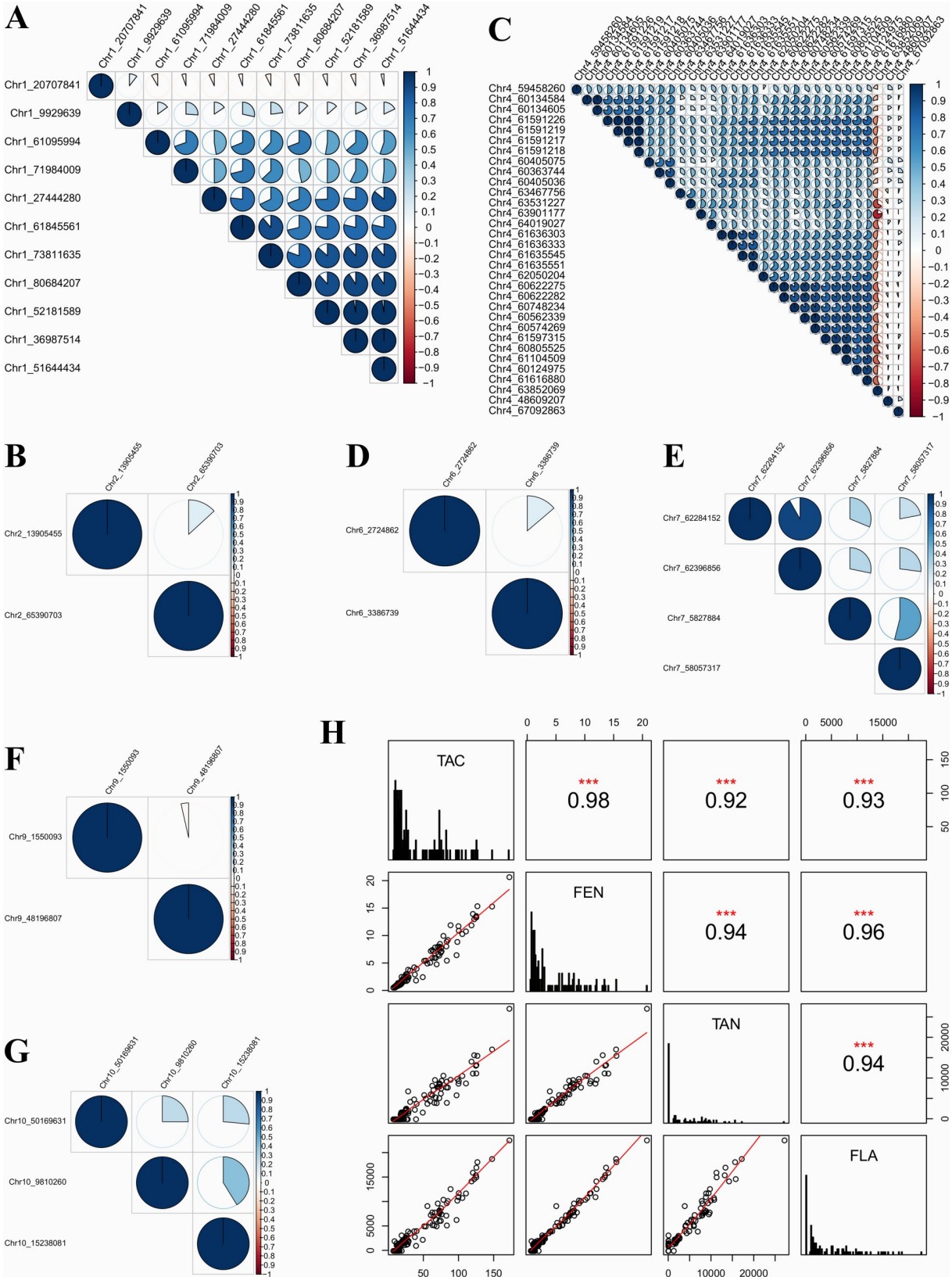

**Fig 3. Pairwise correlation among significant marker loci (A-G) and the relationships between the four antioxidant traits (H).** The filled-in areas of the circles (A-G) show the absolute value of corresponding correlation coefficients. The scale on the right hand side is colored from red (negative correlation) to blue (positive correlation); with the intensity of color scaled 0–100% in proportion to the magnitude of the correlation. Scatter plots (H) with regression lines showing the relationships between the traits are in the lower corner. Correlations between the traits are in the upper corner. Histograms of the mean concentrations of each trait are in the center diagonal. Refer to the text for the description of the traits.

extent of whiskers (Fig 6) was greater in allelic frequencies than in PIC. The reference allele was more frequent than the alternative allele as expected, but the frequency of the alternative allele was higher in SBxSH than in SB, while the frequency of the reference allele was higher in

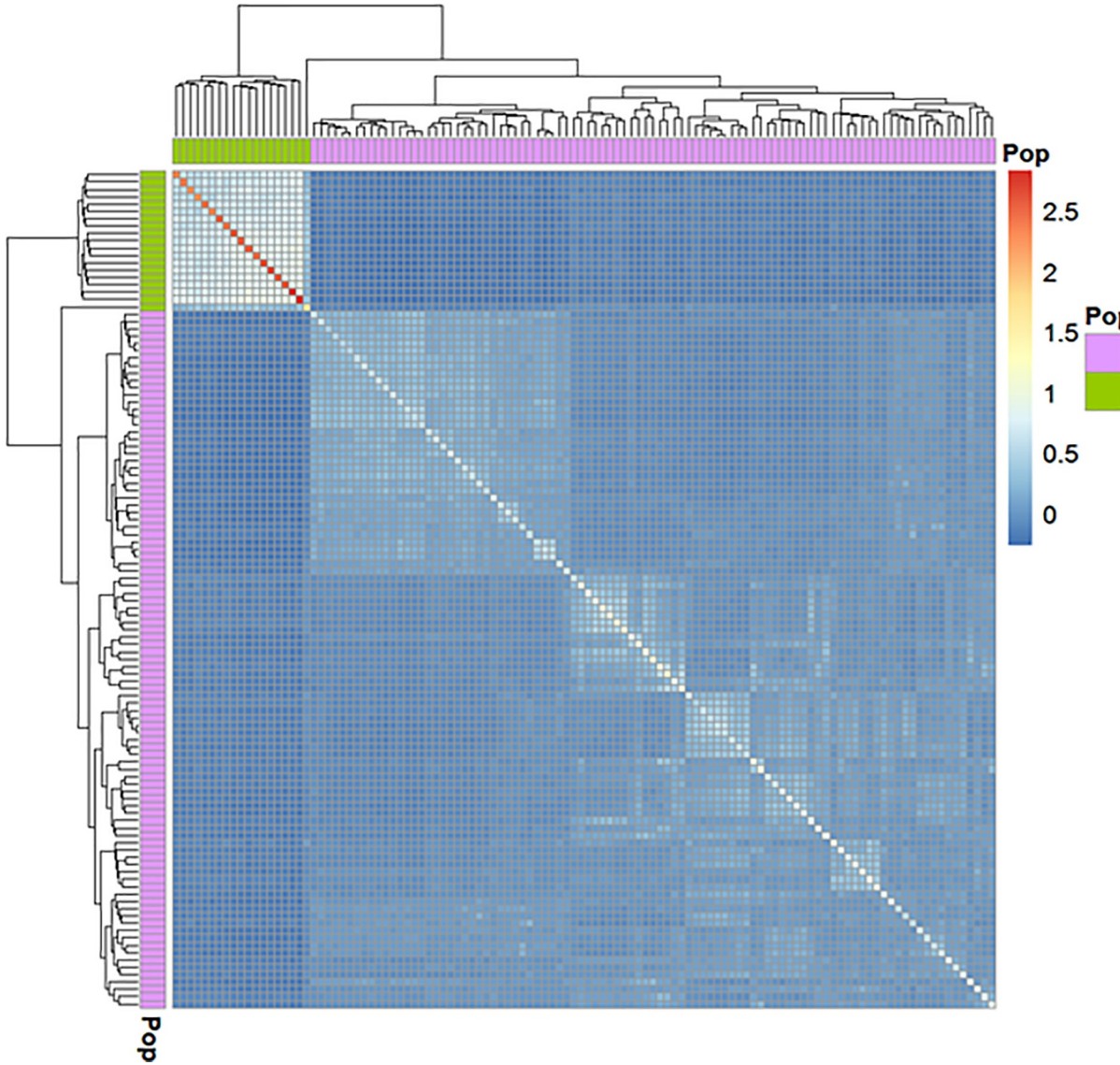

**Fig 4. Sorghum panel genomic relatinship matrix.** Heatmap displaying relationships among the 114 sorghum genotypes used in the GWAS. Pink and green colors identify, respectively, *Sorghum bicolor* and *S. bicolor* × *S. halepense* populations. The white diagonal represents perfect relationship of each genotype with itself; sections of warmer colors in the diagonal represent excess heterozygosity; the symmetric off-diagonal elements represent relationship for pairs of genotypes. The blocks of light colors on the diagonal show clusters of closely related genotypes. The adjoining dendrogram illustrates Kinship groups identified in the sorghum panel.

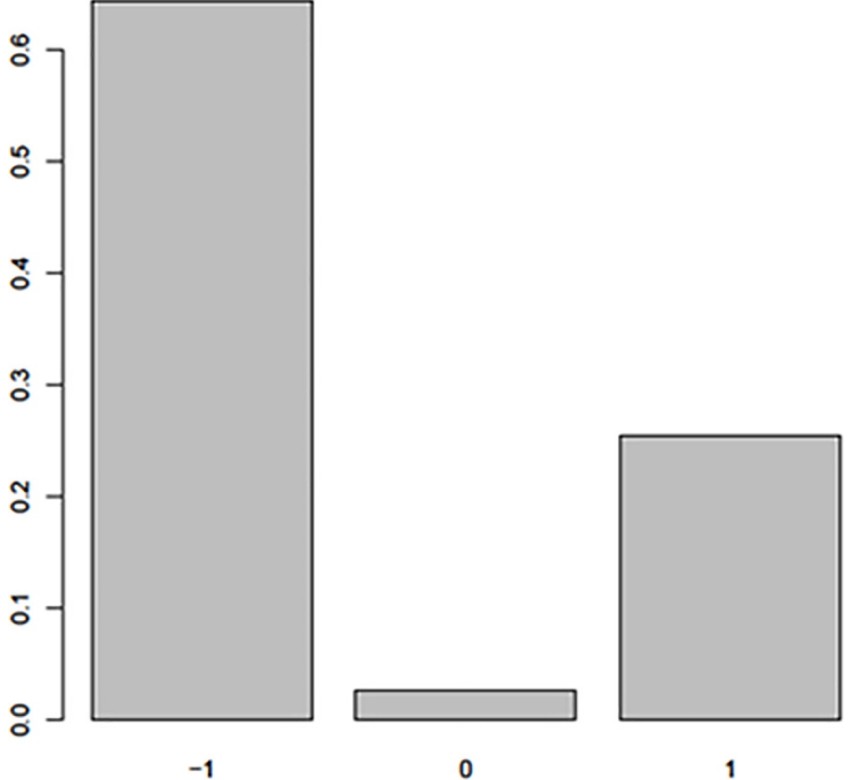

**Fig 5. Genotype frequencies.** Genotypes are coded as -1, 1, 0, respectively, for homozygotes for reference and alternative alleles, and heterozygote.

SB than in SBxSH. Overall, the PIC averaged 0.17 and ranged from 0.02 to 0.5, with SBxSH subpopulation displaying higher (0.23 vs. 0.08) mean PIC value than SB subpopulation.

## Genome-wide association and functional analytics

To investigate the linkage disequilibrium that existed between genetic variants and the antioxidant traits, i.e. total antioxidant activity (TAC), phenolics (FEN), flavonoids (FLA) and condensed tannins (TAN) in the sorghum panel, a whole genome association study was conducted using a genome-wide set of high quality SNP markers and two multi-locus algorithms: SUPER and FarmCPU. Fitness of different GWAS models for all traits was evaluated using Quantile-Quantile (Q-Q) plots (Fig 7) of the observed vs. expected -log10(p) values which should follow a uniform distribution under the null hypothesis. The Q-Q plot showed the -log10(p) values relatively inflated using SUPER, with low scores following the null hypothesis line better when FarmCPU was used. The complete list of significant markers obtained from GWAS analysis is reported in Table 2, whereas the Manhattan plot mapping several strongly associated antioxidant loci is presented in Fig 8. A total of 57 significant SNP loci were detected by the two implemented algorithms across all the traits evaluated: 16 and 44 were identified respectively by FarmCPU and SUPER algorithms, with three markers, Chr1_61095994, Chr4_60363744 and Chr4_61616880, being identified by both methods. Specifically, Chr1_61095994 was significantly associated (hereinafter referred to as associated) with FEN and TAC using FarmCPU, and with TAN using SUPER; Chr4_60363744 was associated with FEN and FLA using SUPER, and with TAC using FarmCPU, while Chr4_61616880

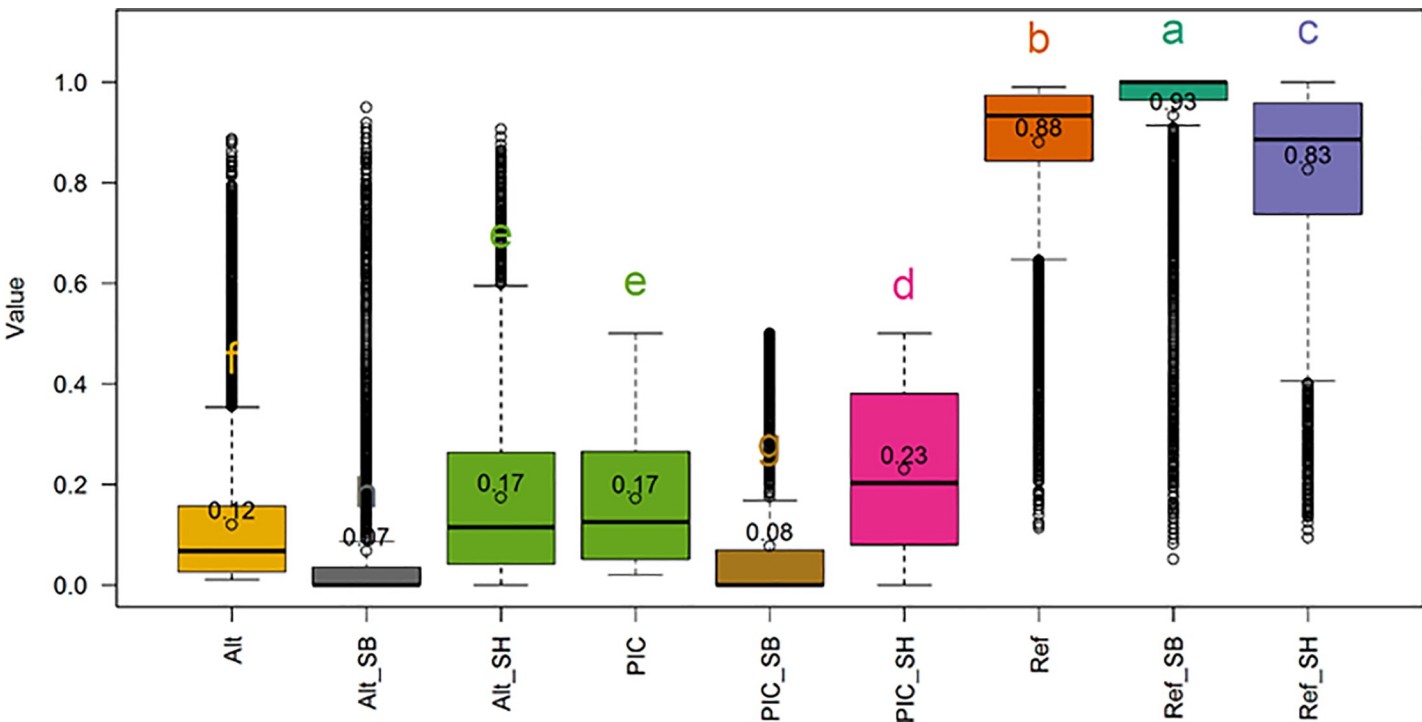

**Fig 6. Alternative (Alt) and reference (Ref) allele frequency and polymorphic information content (PIC) per entire panel and subpopulations.** Open dot inside the rectangles and the numbers inside or outside the rectangles (interquartile ranges) represent, respectively, the position of mean and the mean of the metric of intertest. The horizontal bar inside the rectangles represents the median value. Means with same letter are not significantly different at the 5% level using the Tukey's HSD (honestly significant difference) test.

was associated with FEN, FLA, using SUPER, and with FLA using FarmCPU. Several cases of pleiotropy were observed in which single marker loci were associated with several traits.

Combining unique associations and pleiotropic cases, SUPER and FarmCPU uncovered 79 and 17 associations, respectively, of which 17, 26, 31, and 5 for FEN, FLA, TAC, and TAN using SUPER, and 5, 6, 3, and 3 for FEN, FLA, TAC, and TAN using FarmCPU. Chromosomes 5 and 8 did not show any associations. Chromosome 1 showed associations with all traits except FLA, chromosomes 2 and 10 showed, each, associations for FLA and TAN, chromosomes 3 and 9 for FLA, chromosome 4 was prolific showing most associations for all traits, whereas chromosomes 6 and 7 showed associations for FEN and FLA. Most of the identified markers fell within or in proximity of genomic regions previously reported to harbor QTLs for polyphenol-related traits levels [31–39].

### Functional and major effects markers

Setting the $R^2$ threshold at 15% resulted in 55 associations from 30 significant markers, involving 14 pleiotropic cases in which single markers were associated with 2 to 3 traits. All of these markers had positive effects on respective traits except for two markers on chromosome 9 (Chr9_1550093 and Chr9_48196807) that showed negative effects on flavonoids. These markers that explained at least 15% ($R^2$ ranged from 15 to 31.2%) of the phenotypic variance were therefore considered as major effects markers as suggested by Liu et al. [57], and were highlighted in boldface in Table 2. On the other hand, 2 (Chr9_1550093, Chr10_50169631) (Fig 9) and 30 of the 57 significant markers fell within 'a priori' and 'a posteriori' genes, respectively. One (Chr9_1550093) and twelve (Chr2_13905455, Chr4_60363744, Chr4_60405036, Chr4_60405075, Chr4_61616880, Chr4_63531227, Chr4_63901177, Chr4_64019027,

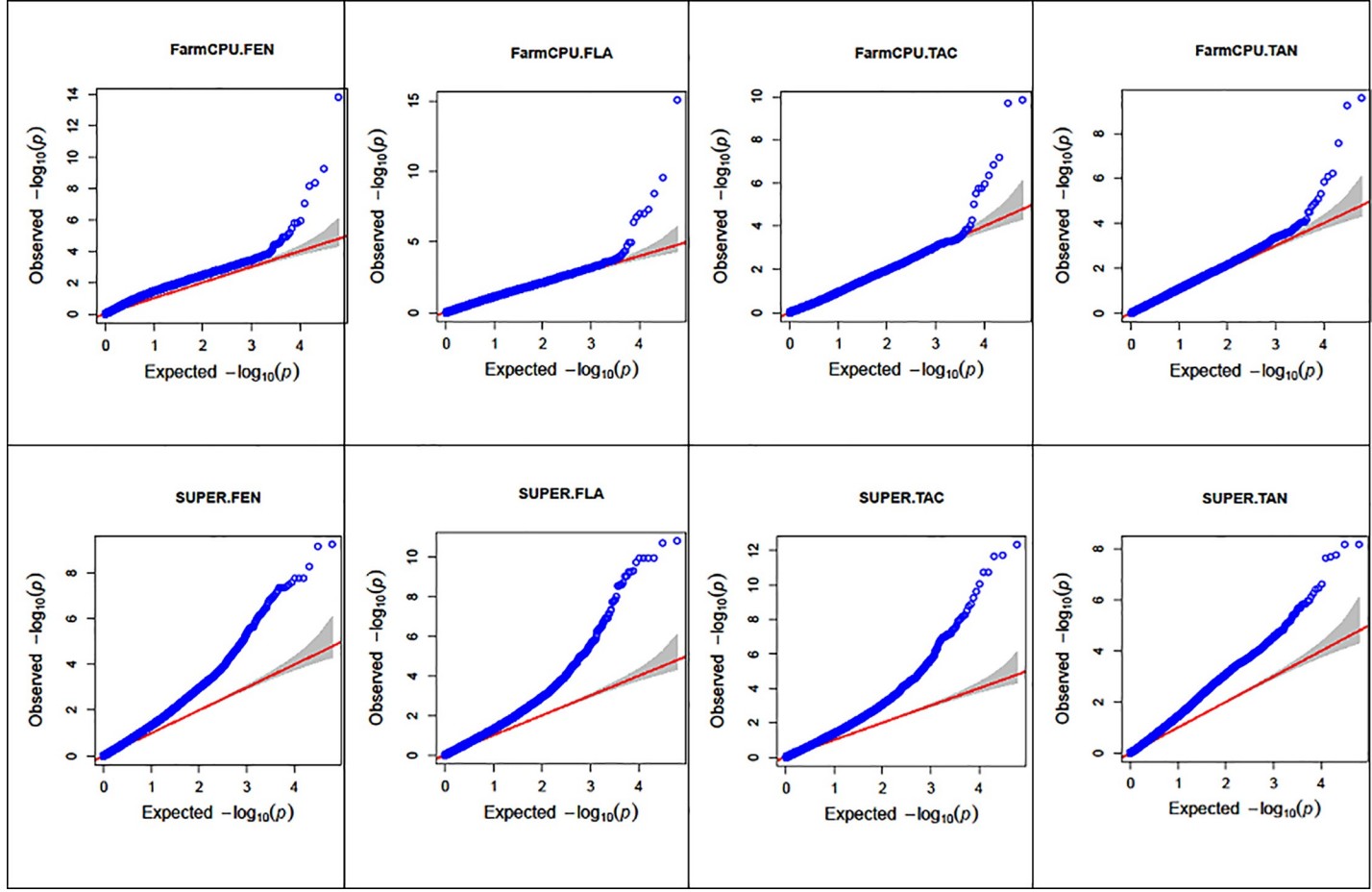

**Fig 7. Quantile-Quantile (QQ) plot of observed against expected probability values (P-values) from the genome-wide association analysis.** TAN, TAC, FEN, FLA, respectively, condensed tannins, total antioxidants, phenols, and flavonoids. Blue circles correspond to the P-values derived from the principal components + kinship model. The red line indicates the expected P-value distribution under the assumption (null hypothesis) that the P-values follow a uniform [0,1] distribution. The dotted lines show the 95% confidence interval for the QQ-plot under the null hypothesis of no association between the SNP and the trait. -$\log_{10}$(P) negative base 10 logarithm of the P-values (probability of type-I error made in GWAS hypotheses testing). Refer to text for the description of the SUPER and FarmCPU algorithms.

Chr7_5827884, Chr7_58057317, Chr7_62284152, Chr9_48196807), respectively, of those within 'a priori' and 'a posteriori' genes were major effects markers (Fig 9, S2 Table). The two markers on chromosome 9 (Chr9_1550093) and 10 (Chr10_50169631) that fell within genes whose putative functions are related to antioxidant activity were therefore considered as functional markers. Functional markers and markers with major effects are the most interesting as far as this work was concerned because they can be usefully targeted for marker-assisted selection schemes in breeding programs. The functional marker on chromosome 10 (Chr10_50169631) and 8 major effect markers (Chr9_1550093, Chr9_48196807, Chr10_9810260, Chr7_5827884, Chr7_58057317, Chr1_61095994, Chr2_13905455, Chr4_48609207) on chromosomes 1, 2, 4, 7, 9 and 10 were novel markers and identified novel QTLs not previously described in other works reported in sorghum QTL Atlas [39].

## Pleiotropic marker loci

Twenty-four cases of pleiotropic markers being associated with 2 to 3 traits were observed using both FarmCPU and SUPER methods. Markers showing pleiotropic effect on three traits

**Table 2.** GWAS results and descriptive statistics of the significant marker loci.

| SNP | MUT | TAC | FEN | TAN | FLA | MODEL | TARGET | P.value | MAF | EFFECT | AAF (SB) | AAF (SB×SH) |
|---|---|---|---|---|---|---|---|---|---|---|---|---|
| Chr1_9929639 | C>T | 11.59 | 10.89 | 10.12 | 9.75 | SUPER | TAC | 3.53E-08 | 0.14 | 15.237 | 0.074 | 0.474 |
| Chr1_20707841 | A>G | 1.00 | 1.61 | 4.41 | 1.36 | FarmCPU | FEN | 4.51E-09 | 0.07 | 2.367 | 0 | 0.389 |
| Chr1_27444280 | C>T | 10.60 | 8.99 | 6.15 | 7.48 | SUPER | TAC | 6.1E-08 | 0.06 | 19.508 | 0.070 | 0 |
| Chr1_36987514 | T>C | 10.70 | 10.17 | 8.08 | 8.01 | SUPER | TAC | 2.13E-08 | 0.05 | 20.124 | 0.066 | 0 |
| Chr1_51644434 | T>A | 11.55 | 10.76 | 9.29 | 8.58 | SUPER | FEN | 4.11E-08 | 0.05 | 2.382 | 0.060 | 0 |
| | | | | | | SUPER | TAC | 9.51E-09 | 0.05 | 21.898 | 0.060 | 0 |
| Chr1_52181589 | G>T | 10.08 | 9.86 | 9.51 | 7.83 | SUPER | TAC | 7.23E-08 | 0.06 | 11.442 | 0.065 | 0 |
| **Chr1_61095994***| C>T | 27.11 | 31.20 | 24.65 | 30.66 | FarmCPU | FEN | 1.43E-14 | 0.10 | 2.964 | 0.120 | 0 |
| | | | | | | FarmCPU | TAC | 6.91E-08 | 0.10 | 23.434 | 0.120 | 0 |
| | | | | | | SUPER | TAN | 1.72E-08 | 0.10 | 3098.641 | 0.120 | 0 |
| Chr1_61845561 | G>T | 9.31 | 9.51 | 8.57 | 7.43 | SUPER | TAC | 4.11E-08 | 0.05 | 16.075 | 0.063 | 0 |
| Chr1_71984009 | G>A | 12.70 | 13.00 | 14.03 | 11.71 | SUPER | TAC | 1.07E-08 | 0.06 | 15.163 | 0.073 | 0 |
| Chr1_73811635 | G>T | 7.59 | 7.69 | 7.16 | 5.87 | SUPER | TAC | 4.28E-08 | 0.06 | 10.835 | 0.068 | 0 |
| Chr1_80684207 | T>C | 12.84 | 13.36 | 9.29 | 11.38 | SUPER | TAC | 9.83E-08 | 0.05 | 23.294 | 0.065 | 0 |
| **Chr2_13905455** | C>A | 9.07 | 12.13 | 20.75 | 11.44 | FarmCPU | TAN | 5.67E-10 | 0.10 | 5710.076 | 0 | 0.588 |
| Chr2_65390703 | G>T | 3.04 | 3.45 | 2.89 | 2.52 | FarmCPU | FLA | 9.69E-08 | 0.11 | 755.402 | 0.071 | 0.278 |
| Chr3_57670375 | A>T | 0.82 | 1.63 | 4.32 | 1.41 | FarmCPU | TAN | 2.84E-08 | 0.10 | -2668.945 | 0 | 0.605 |
| **Chr4_48609207** | C>G | 24.02 | 22.64 | 16.52 | 21.18 | FarmCPU | TAC | 2.03E-10 | 0.09 | 25.743 | 0.096 | 0.079 |
| **Chr4_59458260** | C>G | 24.61 | 25.53 | 19.56 | 25.16 | FarmCPU | FEN | 5.57E-10 | 0.15 | 2.589 | 0.099 | 0.389 |
| **Chr4_60124975***| T>A | 14.58 | 15.00 | 14.13 | 17.85 | SUPER | FEN | 5.32E-10 | 0.11 | 2.603 | 0.095 | 0.211 |
| | | | | | | SUPER | FLA | 1.46E-11 | 0.11 | 3274.460 | 0.095 | 0.211 |
| | | | | | | SUPER | TAC | 2.06E-12 | 0.11 | 23.984 | 0.095 | 0.211 |
| **Chr4_60134584** | C>T | 15.29 | 13.78 | 11.78 | 15.80 | SUPER | TAC | 9.69E-08 | 0.19 | 16.641 | 0.169 | 0.263 |
| **Chr4_60134605** | G>T | 15.29 | 13.78 | 11.78 | 15.80 | SUPER | TAC | 9.69E-08 | 0.19 | 16.641 | 0.169 | 0.263 |
| **Chr4_60363744***| C>G | 19.50 | 19.09 | 16.10 | 20.36 | SUPER | FEN | 3.15E-08 | 0.32 | 1.921 | 0.226 | 0.789 |
| | | | | | | SUPER | FLA | 9.09E-10 | 0.32 | 2343.452 | 0.226 | 0.789 |
| | | | | | | FarmCPU | TAC | 1.38E-10 | 0.32 | 17.992 | 0.226 | 0.789 |
| **Chr4_60405036***| T>C | 21.66 | 21.10 | 20.22 | 22.93 | SUPER | FEN | 7.9E-08 | 0.31 | 1.551 | 0.237 | 0.684 |
| | | | | | | SUPER | FLA | 2.76E-09 | 0.31 | 1995.948 | 0.237 | 0.684 |
| **Chr4_60405075***| C>T | 18.70 | 17.28 | 12.85 | 19.25 | SUPER | FEN | 6.03E-08 | 0.28 | 1.745 | 0.284 | 0.263 |
| | | | | | | SUPER | FLA | 2.71E-09 | 0.28 | 2143.242 | 0.284 | 0.263 |
| Chr4_60562339* | T>A | 9.15 | 10.50 | 9.17 | 13.96 | SUPER | FEN | 1.64E-08 | 0.13 | 2.374 | 0.120 | 0.167 |
| | | | | | | SUPER | FLA | 1.79E-10 | 0.13 | 3168.682 | 0.120 | 0.167 |
| | | | | | | SUPER | TAC | 8.62E-11 | 0.13 | 20.446 | 0.120 | 0.167 |
| Chr4_60574269 | G>T | 12.59 | 14.05 | 12.33 | 17.13 | SUPER | TAC | 1.9E-12 | 0.10 | 25.351 | 0.099 | 0.111 |
| Chr4_60622275* | C>A | 9.48 | 11.42 | 9.67 | 13.97 | SUPER | FEN | 1.67E-08 | 0.14 | 2.703 | 0.127 | 0.211 |
| | | | | | | SUPER | FLA | 5.82E-10 | 0.14 | 3412.285 | 0.127 | 0.211 |
| | | | | | | SUPER | TAC | 1.78E-11 | 0.14 | 23.321 | 0.127 | 0.211 |
| Chr4_60622282* | G>C | 9.48 | 11.42 | 9.67 | 13.97 | SUPER | FEN | 1.67E-08 | 0.14 | 2.703 | 0.127 | 0.211 |
| | | | | | | SUPER | FLA | 5.82E-10 | 0.14 | 3412.285 | 0.127 | 0.211 |
| | | | | | | SUPER | TAC | 1.78E-11 | 0.14 | 23.321 | 0.127 | 0.211 |
| Chr4_60748234* | A>G | 8.39 | 10.21 | 8.99 | 11.47 | SUPER | FEN | 2.71E-08 | 0.08 | 2.831 | 0.055 | 0.158 |
| | | | | | | SUPER | TAC | 3.28E-09 | 0.08 | 22.973 | 0.055 | 0.158 |
| **Chr4_60805525***| A>T | 11.39 | 11.53 | 11.26 | 14.71 | SUPER | FLA | 1.8E-08 | 0.11 | 3406.169 | 0.080 | 0.375 |
| | | | | | | SUPER | TAC | 5.33E-09 | 0.11 | 23.833 | 0.080 | 0.375 |

*(Continued)*

**Table 2.** (Continued)

| SNP | MUT | TAC | FEN | TAN | FLA | MODEL | TARGET | P.value | MAF | EFFECT | AAF (SB) | AAF (SB×SH) |
|---|---|---|---|---|---|---|---|---|---|---|---|---|
| **Chr4_61104509**[*] | A>G | 11.73 | 12.06 | 11.53 | 15.39 | SUPER | FLA | 1.53E-08 | 0.11 | 3377.634 | 0.090 | 0.233 |
| | | | | | | SUPER | TAC | 1.78E-09 | 0.11 | 23.495 | 0.090 | 0.233 |
| **Chr4_61591217**[*] | G>A | 16.56 | 15.12 | 11.61 | 18.20 | SUPER | FEN | 4.41E-08 | 0.12 | 2.603 | 0.094 | 0.211 |
| | | | | | | SUPER | FLA | 1.07E-10 | 0.12 | 3313.476 | 0.094 | 0.211 |
| | | | | | | SUPER | TAC | 6.96E-08 | 0.12 | 24.786 | 0.094 | 0.211 |
| **Chr4_61591218**[*] | T>A | 16.56 | 15.12 | 11.61 | 18.20 | SUPER | FEN | 4.41E-08 | 0.12 | 2.603 | 0.094 | 0.211 |
| | | | | | | SUPER | FLA | 1.07E-10 | 0.12 | 3313.476 | 0.094 | 0.211 |
| | | | | | | SUPER | TAC | 6.96E-08 | 0.12 | 24.786 | 0.094 | 0.211 |
| **Chr4_61591219**[*] | T>C | 16.56 | 15.12 | 11.61 | 18.20 | SUPER | FEN | 4.41E-08 | 0.12 | 2.603 | 0.094 | 0.211 |
| | | | | | | SUPER | FLA | 1.07E-10 | 0.12 | 3313.476 | 0.094 | 0.211 |
| | | | | | | SUPER | TAC | 6.96E-08 | 0.12 | 24.786 | 0.094 | 0.211 |
| **Chr4_61591226**[*] | A>G | 16.56 | 15.12 | 11.61 | 18.20 | SUPER | FEN | 4.41E-08 | 0.12 | 2.603 | 0.094 | 0.211 |
| | | | | | | SUPER | FLA | 1.07E-10 | 0.12 | 3313.476 | 0.094 | 0.211 |
| | | | | | | SUPER | TAC | 6.96E-08 | 0.12 | 24.786 | 0.094 | 0.211 |
| Chr4_61597315[*] | G>A | 12.46 | 12.89 | 11.48 | 17.30 | SUPER | TAC | 4.26E-13 | 0.07 | 29.124 | 0.058 | 0.105 |
| | | | | | | SUPER | TAN | 6.93E-09 | 0.07 | 3571.790 | 0.058 | 0.105 |
| **Chr4_61616880**[*] | G>A | 14.03 | 15.42 | 14.60 | 18.81 | SUPER | FEN | 9.28E-08 | 0.10 | 2.957 | 0.074 | 0.211 |
| | | | | | | FarmCPU | FLA | 8.08E-16 | 0.10 | 3828.922 | 0.074 | 0.211 |
| | | | | | | SUPER | FLA | 1.93E-09 | 0.10 | 3828.922 | 0.074 | 0.211 |
| | | | | | | SUPER | TAC | 1.33E-09 | 0.10 | 26.391 | 0.074 | 0.211 |
| **Chr4_61635545**[*] | C>G | 15.70 | 16.73 | 13.14 | 19.44 | SUPER | FEN | 5.42E-09 | 0.14 | 2.470 | 0.116 | 0.289 |
| | | | | | | SUPER | FLA | 1.91E-11 | 0.14 | 3112.331 | 0.116 | 0.289 |
| | | | | | | SUPER | TAC | 5.62E-10 | 0.14 | 21.877 | 0.116 | 0.289 |
| **Chr4_61635551**[*] | A>G | 12.91 | 13.92 | 11.17 | 16.48 | SUPER | FLA | 9.38E-10 | 0.16 | 2762.077 | 0.116 | 0.395 |
| | | | | | | SUPER | TAC | 7.68E-09 | 0.16 | 19.301 | 0.116 | 0.395 |
| **Chr4_61636303** | A>G | 12.48 | 13.52 | 12.61 | 15.60 | SUPER | FLA | 2.63E-09 | 0.17 | 2689.985 | 0.117 | 0.472 |
| Chr4_61636333 | T>C | 10.94 | 11.93 | 11.20 | 13.99 | SUPER | FLA | 1.76E-08 | 0.18 | 2558.641 | 0.117 | 0.474 |
| Chr4_62050204 | G>A | 8.35 | 8.43 | 9.17 | 11.24 | SUPER | FLA | 2.81E-09 | 0.09 | 2879.601 | 0.073 | 0.158 |
| Chr4_63467756 | T>G | 13.63 | 14.81 | 12.24 | 15.84 | SUPER | TAC | 1.29E-08 | 0.09 | 22.845 | 0.077 | 0.158 |
| **Chr4_63531227**[*†] | A>G | 13.99 | 17.92 | 18.20 | 17.44 | SUPER | FEN | 7.17E-10 | 0.15 | 2.623 | 0.094 | 0.412 |
| | | | | | | SUPER | FLA | 5.32E-10 | 0.15 | 2953.716 | 0.094 | 0.412 |
| | | | | | | SUPER | TAN | 6.54E-09 | 0.15 | 2830.745 | 0.094 | 0.412 |
| Chr4_63852069[*] | G>T | 10.32 | 12.67 | 11.20 | 14.02 | SUPER | FLA | 7.33E-08 | 0.18 | -2509.186 | 0.147 | 0.342 |
| | | | | | | SUPER | TAC | 2.85E-08 | 0.18 | -17.235 | 0.147 | 0.342 |
| **Chr4_63901177** | C>G | 15.35 | 18.53 | 16.36 | 19.69 | SUPER | TAC | 6.14E-09 | 0.16 | 15.937 | 0.126 | 0.333 |
| **Chr4_64019027** | G>C | 21.64 | 24.79 | 27.68 | 25.75 | SUPER | TAC | 2.34E-10 | 0.22 | 19.808 | 0.120 | 0.711 |
| Chr4_67092863 | T>A | 12.09 | 9.10 | 7.74 | 8.54 | FarmCPU | FLA | 5.33E-08 | 0.07 | 3127.442 | 0.054 | 0.158 |
| Chr6_2724862 | T>C | 7.62 | 8.14 | 5.60 | 11.21 | FarmCPU | FLA | 2.85E-10 | 0.05 | 3132.992 | 0.066 | 0 |
| Chr6_3386739 | A>G | 2.39 | 2.25 | 3.11 | 2.32 | FarmCPU | FEN | 7.15E-09 | 0.10 | 1.481 | 0.049 | 0.353 |
| **Chr7_5827884** | A>C | 18.02 | 18.42 | 20.97 | 18.36 | FarmCPU | FEN | 9.16E-08 | 0.23 | 1.513 | 0.159 | 0.579 |
| **Chr7_58057317** | A>G | 16.81 | 18.50 | 22.32 | 20.30 | SUPER | FLA | 1.04E-08 | 0.24 | 1797.672 | 0.147 | 0.684 |
| **Chr7_62284152**[*] | A>G | 17.84 | 21.00 | 16.38 | 22.10 | SUPER | FEN | 1E-07 | 0.10 | 2.633 | 0.122 | 0 |
| | | | | | | SUPER | FLA | 4.36E-08 | 0.10 | 3123.129 | 0.122 | 0 |
| | | | | | | SUPER | TAN | 2.32E-08 | 0.10 | 2509.794 | 0.122 | 0 |
| **Chr7_62396856**[*] | A>G | 17.33 | 20.20 | 17.10 | 22.39 | SUPER | FLA | 2E-08 | 0.10 | 3179.647 | 0.119 | 0 |
| | | | | | | SUPER | TAN | 2.08E-08 | 0.10 | 2762.032 | 0.119 | 0 |

(*Continued*)

**Table 2.** (Continued)

| SNP | MUT | TAC | FEN | TAN | FLA | MODEL | TARGET | P.value | MAF | EFFECT | AAF (SB) | AAF (SB×SH) |
|---|---|---|---|---|---|---|---|---|---|---|---|---|
| **Chr9_1550093[T]** | C>G | 10.72 | 13.02 | 9.83 | 16.80 | SUPER | FLA | 7.83E-08 | 0.10 | -3250.146 | 0.095 | 0.100 |
| **Chr9_48196807** | C>G | 16.58 | 19.83 | 18.52 | 16.69 | FarmCPU | FLA | 9.71E-08 | 0.38 | -1517.948 | 0.310 | 0.667 |
| **Chr10_9810260** | A>G | 9.88 | 12.01 | 15.89 | 12.77 | FarmCPU | TAN | 2.58E-10 | 0.23 | 2376.441 | 0.140 | 0.658 |
| Chr10_15238081 | T>C | 8.75 | 9.19 | 8.94 | 8.75 | FarmCPU | FLA | 3.8E-09 | 0.20 | 818.810 | 0.156 | 0.421 |
| Chr10_50169631[T] | A>G | 10.81 | 11.34 | 10.00 | 10.98 | SUPER | FLA | 5.04E-08 | 0.08 | 2216.985 | 0.095 | 0 |

*, [T] Pleiotropic effects and functional markers, respectively; in bold: major effect markers

included Chr1_61095994 displaying effect on FEN, TAC, and TAN, Chr4_60124975, Chr4_60363744, Chr4_60562339, Chr4_60622275, Chr4_60622282, Chr4_61591217, Chr4_61591218, Chr4_61591219, Chr4_61591226, Chr4_61616880, and Chr4_61635545 on FEN, FLA, and TAC, Chr4_63531227 and Chr7_62284152 on FEN, FLA, and TAN. The following markers were associated, each, with two traits: Chr1_51644434, Chr4_60748234 with FEN and TAC, Chr4_60405036 and Chr4_60405075 with FEN and FLA, Chr4_60805525, Chr4_61104509, Chr4_61635551, and Chr4_63852069 with FLA and TAC, Chr7_62396856 with FLA and TAN, and Chr4_61597315 with TAC and TAN. Interestingly, we did not observe a marker with pleiotropic effect on the four antioxidant traits evaluated in this study.

## Classes of candidate genes in LD with major effects markers

The genomic regions harboring major effects markers were scanned for the presence of possible candidate genes considering 500 Kb upstream and downstream from the SNP position as suggested by the LD decay analysis. The complete list of candidate genes in proximity of major effects markers is reported in S1 Table. A total of 61 candidate genes were identified and classified in appropriate categories based on their putative antioxidant functions: biosynthesis, regulation, transport and oxidation. Most of the genes (26) found belonged to the transport class followed by those (21) involved in the regulation, biosynthesis (9), and in the oxidation (5). Only two markers (Chr4_48609207 and Chr4_63531227) were not found in the vicinity of candidate genes within the 500 Kb LD limit. The nearest candidate genes for these markers were observed by enlarging the distance to 809 and 729 Kb from the respective positions of the markers. The biggest class of candidate genes was comprised of 20 genes similar to Glutathione-S-transferase (GST). Nineteen of these genes were on Chr1, and 1 on Chr9. In *Zea mays*, GST, also called bronze2 (bz2), encodes for a GST protein transporting anthocyanins into the vacuole [58]. In the regulation class were found genes similar to MYB transcription factor (13 genes), WD40 genes (5 genes) and 1 gene similar to Basic helix-loop-helix (BHLH). These three genes together form a protein complex (MBW) that have been studied to be involved in the flavonoid biosynthesis [59]. We came across 3 putative genes similar to peroxidase which encodes for a protein known to catalyze the oxidation of phenolic substrates; peroxidases are also able to produce ROS through the hydroxylic cycle [52]. Two genes were annotated as similar to a Multidrug resistance protein (MRP), one on Chr1 and one on Chr10. In *Zea mays* a MRP gene was shown to be involved in the transport of anthocyanins [60].

## Total antioxidant capacity (TAC)

For total antioxidant capacity (TAC) both FarmCPU and SUPER methods found significant markers on chromosomes 1 and 4, all of which, except Chr4_63852069, displayed positive

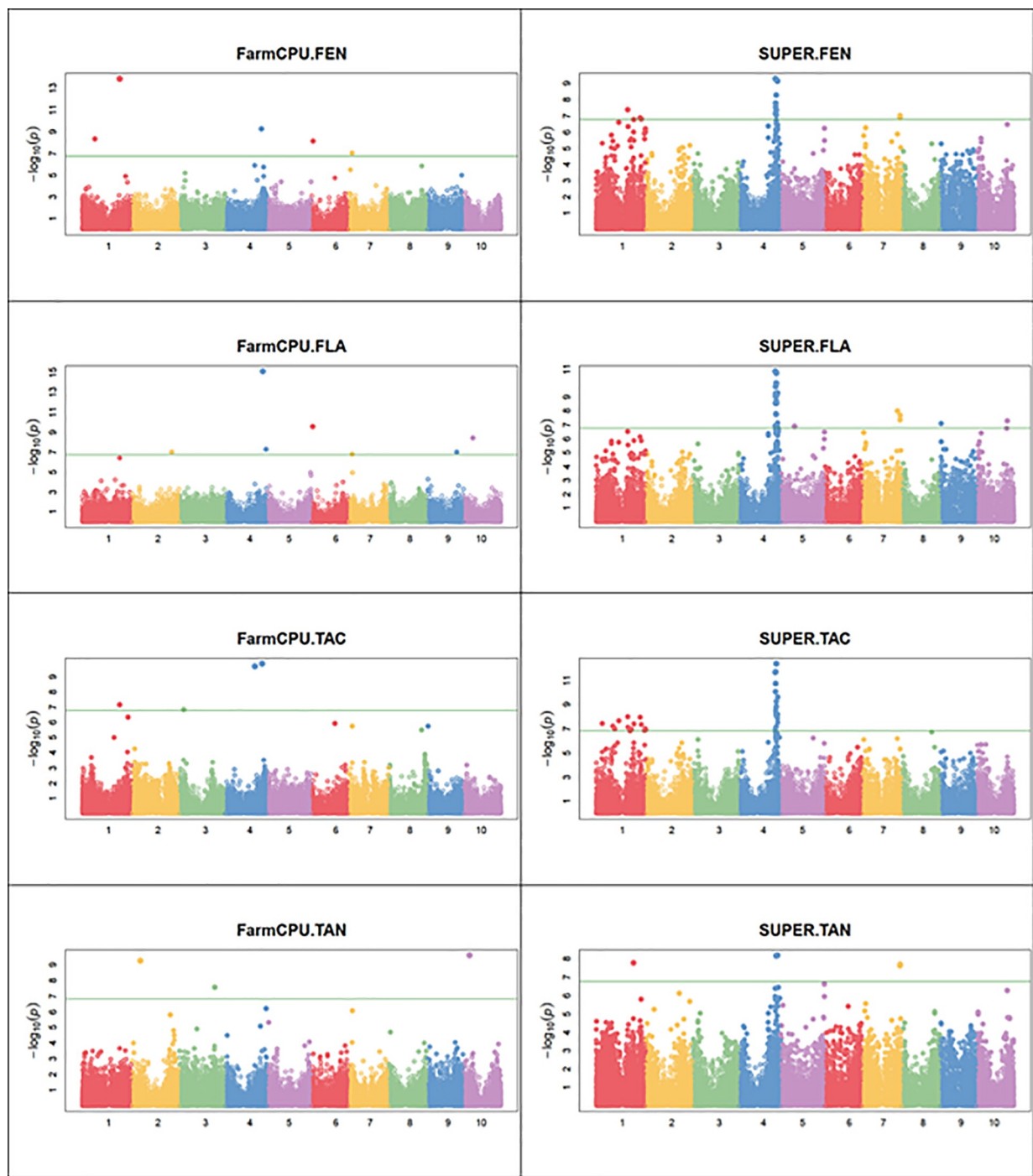

**Fig 8. Manhattan plots representing several strongly associated antioxidant loci for the antioxidant traits.** Each circle in the scatter plot represents a SNP, with the X-axis showing genomic location. Numbers 1 to 10 on X-axis represent the ten *Sorghum bicolor chromosomes*. The Y-axis shows the association level: -log10(P) is the negative base 10 logarithm of the P-values (probability of type-I error made in GWAS hypotheses testing). The solid horizontal line represents the genome-wide significance threshold as explained in the text. Regions with −log10 p-values above the threshold are candidates as in Table 2. Each plot shows the output of an algorithm for a specific target trait in the form "algorithm.trait". TAN, TAC, FEN, FLA, respectively, condensed tannins, total antioxidants, phenols, and flavonoids. Refer to text for the description of the SUPER and FarmCPU algorithms.

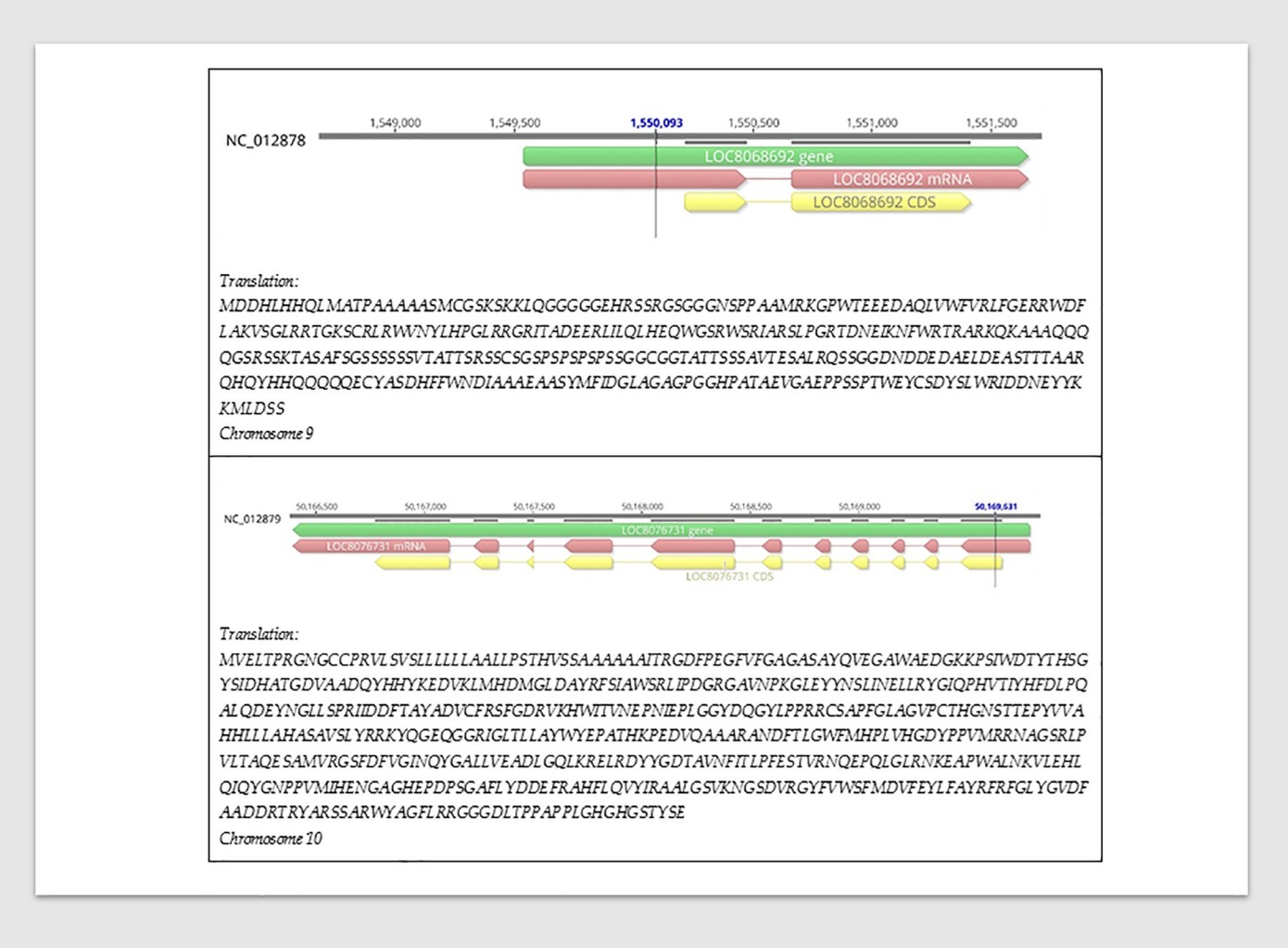

**Fig 9. Functional markers, genes, coding sequence, transcripts, and the aminoacid sequences of the gene products of interest.** The position (in base pairs) of the functional marker is indicated by a vertical bar. For each chromosome, from top to bottom: numbers represent the physical map (distance) in base numbers; solid line represents the region of the chromosome of interest as identified by unique NCBI identification number (ID) for the reference Sorghum bicolor NCBIv3 primary assembly; dashed horizontal lines are exons; horizontal green arrowed-bar represents the gene locus (and ID) of interest derived by automated computational analysis using Gnomon eukaryotic gene prediction algorithm; horizontal dashed red arrowed-bar represents the messenger ribonucleic acid macromolecule (along with ID) corresponding to gene of interest; the horizontal dashed yellow arrowed-bar represents the coding sequence (CDS along with ID) of the gene of interest. The direction of arrows indicates the DNA and RNA 5′-to-3′ direction. The translation reports the letters representing the sequence (in one-letter code format) of the aminoacids making up the gene product of interest.

effect on the trait of interest. FarmCPU identified Chr1_61095994, Chr4_48609207 and Chr4_60363744 markers. SUPER method identified a total of 31 SNPs, different from those found with FarmCPU method: 22 on chromosome 4 (between 60.1 Mb and 64 Mb) and 9 located in different positions on Chr1; these nine markers on Chr1 were found monomorphic for the reference allele in the SBxSH population (Table 2).

Seventeen major effects ($R^2$ 15–27%) markers were identified, all with a positive effect on TAC. These associations involved one marker (Chr1_61095994) on chromosome 1 and 16 markers (Chr4_48609207, Chr4_60124975,Chr4_60134584, Chr4_60134605, Chr4_60363744, Chr4_60805525, Chr4_61104509, Chr4_61591217, Chr4_61591218, Chr4_61591219, Chr4_61591226, Chr4_61616880, Chr4_61635545, Chr4_61635551, Chr4_63901177,

Chr4_64019027) on chromosome 4. Chr1_61095994 ($R^2$ 27.1%) was identified using Farm-CPU and was monomorphic for the reference allele in the SBxSH population. It is worth noticing that this marker is located in the vicinity (343–494 Kb) of 19 genes similar to GST (Glutathione-S-transferase) family; 14 of these 19 genes were annotated by Rhodes et al. [31,37] to be homologous to *Zea mays* GRMZM2G016241 gene. The gene nearest (234 Kb) to Chr1_61095994 is Sobic.001G320900.1 similar to a Multidrug resistance protein (MRP).

On Chr4 were found a region dense with significant SNPs located in different positions from 60.1 to 64 Mb. The first part of the region delimited by markers Chr4_60124975 and Chr4_60363744, harbors two genes similar to a WD40 (Sobic.004G256500.1 and Sobic.004G257400.1). The following part of the region is between Chr4_60805525 and Chr4_61635551 markers, where are located 6 putative genes: 3 genes involved in the regulation, two of which are similar to MYB (Sobic.004G267000.2 and Sobic.004G270600.1), another gene similar to Basic helix-loop-helix (BHLH) transcription factor (Sobic.004G270900.3), 1 gene similar to Leucoanthocyanin reductase (LAR) (Sobic.004G267800.1) that is involved in the biosynthetic pathway of polyphenols, 1 gene similar to 9-cis-epoxycarotenoid dioxygenase / beta carotene dioxygenase (Sobic.004G268500.1) that encodes for a protein that cleaves the reaction of beta-carotene at the central bound into two molecules of retinal, modulating beta-carotenoid dietary function [61], and 1 gene (Sobic.004G270800.3) similar to WD40. In the proximity of the two markers Chr4_63901177 and Chr4_64019027 (at about 200 Kb) were found another two genes (Sobic.004G303400.1 and Sobic.004G303600.1) similar to MYB belonging to the regulation category.

## Total condensed tannins (TAN)

The GWAS analysis for total condensed tannins produced a different output using the two methods (Table 2). SUPER identified 5 significant markers located on chromosomes 1, 4 and 7, all having a positive effect on the trait. In particular, three markers (Chr1_61095994, Chr7_62284152, Chr7_62396856) were polymorphic only in the SB population. FarmCPU method, on the other hand, yielded 3 significant SNPs on chromosomes 2, 3 and 10, two (Chr2_13905455 and Chr10_9810260) and one (Chr3_57670375) with positive and negative effect, respectively. Interestingly, the markers on chromosomes 2 and 3 were polymorphic only in SBxSH population (Table 2).

Six major effects markers ($R^2$ = 16–25%) were identified on chromosomes 1 (Chr1_61095994), 2 (Chr2_13905455), 4 (Chr4_63531227), 7 (Chr7_62284152, Chr7_62396856) and 10 (Chr10_9810260). Chr1_61095994 showed major effect also on TAC and the relative functional analysis was described above. Chr2_13905455 mapped 332 Kb from Sobic.002G115700.1 similar to Putative chalcone synthase, CHS, homolog to a TT4 gene in *Arabidopsis thaliana* (AT5G13930). We couldn't find a candidate gene in LD with Chr4_63531227 but, scanning the genome farther away (729 Kb) from this SNP, we came across a gene annotated as similar to Zn-finger transcription factor. This gene is homologous to TT1 in A. thaliana (AT1G34790) [31,37], and interestingly, it is located in the beginning of the metabolic pathway of tannins. Chr7_62284152 mapped 374 Kb from Sobic.007G186200.1 gene, annotated as similar to a putative anthocyanin-related membrane protein 1 that encodes for a protein involved in the transport of anthocyanins. Chr7_62396856 was found in LD with 3 putative genes Sobic.007G192300.1, Sobic.007G193300.1, and Sobic.007G193900.1 which are similar, respectively, to peroxidase, MADS-box transcription factor homolog to TT16 in A. thaliana (AT5G23260), and WD40 repeat protein encoding genes. Finally, on Chr10 marker Chr10_9810260 is located close to a gene (Sobic.010G106601.1) encoding for a MYB DNA-binding protein.

## Total phenolics (FEN)

SUPER and FarmCPU identified, respectively, 17 and 5 SNPs associated to total phenolics (FEN), covering chromosomes 1, 4, 6 and 7 and all having a positive effect on the trait (Table 2). Out of these 22 significant markers, 16 (15 with SUPER and 1 with FarmCPU) are located in the region between 59 and 64Mbp on chromosome 4. FarmCPU, on the other hand, identified an association involving chromosome 1 (marker Chr1_61095994) and accounting for the highest amount ($R^2 = 31.20\%$) of the FEN variance observed in the panel evaluated in this work. One significant marker on chromosome 1 (Chr1_20707841) showed polymorphism only in the SBxSH subgroup, as the alternative allele was not detected in the SB population (Table 2).

Overall, fifteen major effects markers ($R^2$ 15–31%) were identified by the two GWAS methods of which 1 (Chr1_61095994) on chromosome 1, 12 (Chr4_59458260, Chr4_60124975, Chr4_60363744, Chr4_60405036, Chr4_60405075, Chr4_61591217, Chr4_61591218, Chr4_61591219, Chr4_61591226, Chr4_61616880, Chr4_61635545, Chr4_63531227) on chromosome 4, and 2 (Chr7_5827884, Chr7_62284152) on chromosome 7. Markers Chr1_61095994, Chr4_60124975, Chr4_60363744, Chr4_61591217, Chr4_61591218, Chr4_61591219, Chr4_61591226, and Chr4_61635545 were also associated with TAC and the putative genes found near these markers were described above. Similarly, putative genes near Chr4_63531227 and Chr7_62284152 were described under the TAN trait above. Chr4_59458260 mapped 118 Kb from a MYB transcription factor (Sobic.004G242900.1), 356 Kb from Sobic.004G248700.1 similar to a WD40 and 379 Kb from Sobic.004G242600.1 gene similar to a peroxidase.

## Total flavonoids (FLA)

The total flavonoid content (FLA) was found to be associated to 31 markers, 26 of which identified by the SUPER algorithm and 6 by FarmCPU, with one (Chr4_61616880) being common to the two methods. The latter marker was the most relevant based on FarmCPU ($R^2 = 18,81$), while SUPER showed Chr4_60405036 as explaining the highest percentage ($R^2$ 22.93%) of the observed FLA variance, followed by Chr7_62284152 and Chr7_62396856 ($R^2$ 22%). Again, chromosome 4 contained most of the significant markers (23 out of 31).

Twenty major effect markers ($R^2$ 15–23%) were identified by SUPER and FarmCPU of which 15 (Chr4_60124975, Chr4_60363744, Chr4_60405036, Chr4_60405075, Chr4_60805525, Chr4_61104509, Chr4_61591217, Chr4_61591218, Chr4_61591219, Chr4_61591226, Chr4_61616880, Chr4_61635545, Chr4_61635551, Chr4_61636303, Chr4_63531227) on chromosome 4, 3 (Chr7_58057317, Chr7_62284152, Chr7_62396856) on chromosome 7 and 2 (Chr9_1550093, Chr9_48196807) on chromosome 9. The two SNPs on chromosome 9 were the only showing a negative effect on the trait. Most of these major effect markers expressed pleiotropic effects and the putative genes near which they mapped were described above. In particular, Chr4_60124975, Chr4_60363744, Chr4_61104509, Chr4_61591217, Chr4_61591218, Chr4_61591219, Chr4_61591226, Chr4_61635545 was also associated with FEN and TAC, Chr4_63531227, Chr7_62284152 with FEN and TAN, Chr4_60405036, Chr4_60405075, Chr4_61616880 with FEN, whereas Chr7_62396856 was also associated with TAN.

Among the remaining major effect markers, Chr7_58057317 mapped at 75 and 83 Kb from Sobic.007G149000.1 and Sobic.007G148900.1 genes, both found to be similar to Flavone 3'-hydroxylase (F3'H) and homologous to TT7 in A. thaliana (AT5G07990). At 415 Kb distance from the marker is located Sobic.007G146500.1 gene similar to MYB gene.

Chr9_1550093 mapped within the 2114 bp long Sobic.009G016600.1 gene (1,549,539–1,551,652 bp) which is similar to MYB gene, encoding for a DNA binding protein (Fig 9). This

marker was also found in LD with other 'a priori' genes: a MYB category gene (Sobic.009G016633.1), a gene similar to H+-ATPase proton pump (Sobic.009G020900.1), and another similar to GST (Sobic.009G023050.1). The marker Chr10_50169631 mapped within the 3395 bp long (50,169,787–50,166,393 bp) Sobic.010G170300.1 putative gene annotated as similar to Beta-glucosidase (Fig 9). It was discovered using FLA as target trait, showed positive effect and explained 11% of the variance observed on this trait. A second gene (Sobic.010G169000.1) was found at 271 Kb from the marker, and was similar to MRP anthocyanin transporter and homolog to ZmMRP3ZmMRP3 gene in *Zea mais* (GRMZM2G111903).

### Pairwise statistical association among GWAS significant SNPs marker loci

Among the significant markers, no negative correlation was observed (Fig 3A–3G). The significant markers on chromosomes 2, 6, 9, and 10 were statistically unrelated with a pairwise Pearson coefficient $r \leq 0.5$. On Chromosome 1, 34 pairwise correlations were high ($r \geq 0.5$). Markers Chr1_20707841and Chr1_9929639 were unrelated to the remaining 9 (Chr1_61095994, Chr1_71984009, Chr1_27444280, Chr1_61845561, Chr1_73811635, Chr1_80684207, Chr1_52181589, Chr1_36987514, and Chr1_51644434). Among these nine markers, the pairwise correlations were above 0.5 except the pairs Chr1_61095994/ Chr1_27444280 and Chr1_71984009/ Chr1_80684207. The pairwise correlation was perfect ($r \approx 1$) among three markers Chr1_52181589, Chr1_36987514, and Chr1_51644434 located within an interval of about 15.19Mb. On Chromosome 4, markers Chr4_48609207 and Chr4_67092863 were unrelated with any other markers. Marker Chr4_63852069 showed negative correlation coefficients with the rest of the markers, but pairwise r was $\leq -0.5$ only between this marker and Chr4_61597315, Chr4_60805525, Chr4_61104509, Chr4_60124975, Chr4_61616880, Chr4_60562339, Chr4_60622275, Chr4_60622282, Chr4_63467756, Chr4_63531227, Chr4_63901177, Chr4_64019027.

Three blocks of highly ($r \geq 0.7$) correlated markers were observed on chromosome 4. Markers in the first block (Chr4_61591226, Chr4_61591219, Chr4_61591217, and Chr4_61591218) were perfectly correlated ($r = 1$) with each other and were located within 9 base pairs (bp) of each other. The second block was made up of four markers (Chr4_61636303, Chr4_61636333, Chr4_61635545, and Chr4_61635551) located within 758 bp from each other. Of these markers, a perfect correlation was observed between marker pairs Chr4_61635545/Chr4_61635551 and Chr4_61636303/Chr4_61636333. The third block comprised 10 markers (Chr4_60622275, Chr4_60622282, Chr4_60748234, Chr4_60562339, Chr4_60574269, Chr4_61597315, Chr4_60805525, Chr4_61104509, Chr4_60124975, and Chr4_61616880) located within an interval of about 1.49 Mb. Block 1 was highly (but not perfectly) correlated with block 3, and the two blocks were located within a 1.47 Mb interval.

## Discussion

Sorghum breeding programs rely heavily on phenotypic selection to control the contents of polyphenols in the grains. The use of grain color as a proxy to polyphenols concentration was nonetheless complicated by the necessity to have a variety of information including pericarp thickness, pigmented testa, spreader genes, and endosperm appearance, which are correlated with the production of sorghums with increased phenols and antioxidant activity levels [62–66]. Using marker assisted breeding can simplify and expedite breeding for antioxidant activity. In this work, we used genome-wide association study (GWAS) to discover useful variants in linkage disequilibrium with genetic factors controlling antioxidant traits in grain sorghum. GWAS is an important alternative for mapping quantitative traits, and it is implemented extensively in crop improvement programs.

The broad-sense heritability of the traits (FEN, FLA, TAC, TAN) evaluated in this work was very high (Table 1), indicating that the phenotypic variations of the measured antioxidant metrics are mainly affected by genetic factors, and therefore this panel can be used for genetic linkage disequilibrium assessments. On the other hand, the pairwise correlation among the four antioxidant traits (Fig 2) was positively very high implying that the proportion of variance shared by the traits evaluated was mostly due to genetic causes. As a corollary to this statement, the observed phenotypic correlation can be considered as a measure of existing overlap between the sets of genetic influences on respective antioxidant measurements but, cannot be considered as a measure of the absolute magnitudes of those influences. On the other hand, it can be inferred that the observed high correlation between antioxidant components and the TAC is an indication that the antioxidant components evaluated in this work were major source of antioxidant activity in sorghum grain.

A perfect correlation between traits implies that genetic influences on the traits of interest are identical, and this situation can be explained by the existence of either pleiotropy (causal overlap), linkage disequilibrium, or ascertainment bias as related to sampling bias. Although we cannot totally ignore the possibility of ascertainment bias in virtue of the small size of the sample used in this work, we nonetheless witnessed extensive cases of markers with pleiotropic effects on the phenotype (Table 2), and cases of blocks of markers displaying high pairwise correlation (Fig 3A–3G), meaning that these two phenomena effectively explained the observed correlation among traits. With regard to the size of the sample used in this study, since the SBxSH lines and *S. bicolor* landraces evaluated have a long history spanning, respectively, more than 6 and 20 years, they have accumulated cycles of meiotic and recombination events that qualify the sample size used herein for GWAS investigations [9,14,67]. In addition, the population size did not show major data quality concerns as demonstrated by the uncovered associations that were significant and consistent with previous reports.

The Q-Q plot was used in this work to assess the fitness of the GWAS models and showed the -log10(p) values relatively inflated using the SUPER approach (Fig 7) The same Q-Q pattern was reflected in recent works [68]. Since the SUPER method was ranked among the most statistically powerful algorithms, and given that the family and population structures were properly controlled in this work using kinship (G matrix) and principal components, it can be inferred that the light offset observed in lower SUPER scores were probably caused by the high sensibility and resolution of this method, that in particular, resulted in more significantly linked markers on chromosome 4 (Fig 8). The two GWAS approaches implemented in this work uncovered different sets of significant markers, with only three markers identified in common. Commonly identified markers are particularly interesting but, as Xu et al. [50] pointed out, different methods can detect different markers due for instance to differing sensitivity to minor allele frequency x effect sizes combinations.

Among the previously discovered antioxidant genes, it is worth mentioning that Tannin1 which was found at 262 Kb from the nearest marker Chr4_62050204 did show major effect, and this is agreement with Rhodes et al. [31,37]. Yellow seed1 was located on chromosome1 but was not included in the QTLs reported herein as it was at 3.6 Mb far away from the nearest Chr1_71984009 marker. On the other hand, marker Chr7_58057317 on chromosome 7 showed major effect on FLA and mapped at 75 and 83 Kb from Sobic.007G149000.1 and Sobic.007G148900.1 genes both found to be similar to Flavone 3'-hydroxylase (F3'H) and homologous to TT7 in A. thaliana (AT5G07990), a gene directly involved in flavonoid biosynthetic process [69].

The GWAS investigations reported herein were based on phenotypic data collected from two open field trials conducted over two years. Two-year trials are a threshold standard in agricultural research. Several studies (e.g., [31,37,50]) similar to ours reported GWAS findings

based on one-year trial. Sorghum antioxidants concentration is a highly heritable trait, and indeed heritability in this work was above 0.9, meaning that the environmental noise for these qualitative traits is negligible and could not significantly affect the statistical inferences on which we based our findings.

### Novel major effect SNPs, functional SNPs, gene hotspots

Previous GWAS works reported molecular markers most of which with small effects on anti-oxidants levels [31–38]. In the present study, 55 associations from 30 major effects ($R^2 \geq 15\%$) markers were reported, among which 14 pleiotropic cases were observed involving 2 to 3 traits. These major effects SNPs are expected to effectively boost genetic gain of the traits of interest per unit of time and cost. The negative and positive effects markers can allow breeders to conduct divergent breeding for antioxidant levels. On the other hand, 8 novel major effects markers were discovered in this work and can be used for new causal gene discovery. Two functional novel markers were identified at the proximal and distal sides, respectively, of chromosomes 9 (Chr9_1550093) and 10 (Chr10_50169631). These novel markers characterize novel quantitative traits loci and can be directly used in new breeding applications to increase sorghum antioxidants concentrations quantitatively.

On chromosome 1, marker Chr1_61095994 was found significant for FEN and TAC using FarmCPU method, and for TAN using SUPER, and can be considered as of major interest in sorghum breeding for antioxidant concentration. Indeed, this marker showed high $R^2$ (25%) for TAN, and the highest $R^2$ values of 27.1% and 31.2%, respectively, for TAC and FEN. Fur-thermore, this marker is located in proximity of a chromosome 1 hotspot of 19 genes that are all similar to GST gene which, in *Zea mays*, is also called bronze2 (bz2) and encodes for a pro-tein conjugating anthocyanins into the vacuole [58]. Another chromosomal hotspot region is represented by the interval 60.1 to 64 Mb on chromosome 4 which is dense with significant SNPs located in different positions from 60.1 to 64 Mb. This region can be a good target for marker assisted breeding because it harbors genes similar to WD40 (Sobic.004G256500.1, Sobic.004G257400.1, and Sobic.004G270800.3), MYB (Sobic.004G267000.2 and Sobic. 004G270600.1), Basic helix-loop-helix transcription factor (Sobic.004G270900.3), Leucoantho-cyanin reductase (Sobic.004G267800.1), 9-cis-epoxycarotenoid dioxygenase / beta carotene dioxygenase (Sobic.004G268500.1). These genes are involved in the regulation, biosynthesis, and oxidation of the polyphenols.

### Novel contribution of *S. halepense* genome to antioxidant variability

Although *Sorghum halepense* is crossed to domesticated sorghum with the main target of introducing perenniality, our findings showed that its gene pool harbors many alleles useful for improving several other traits [10] including antioxidant properties of the grain. As expected [20,30], SBxSH lines outperformed *S. bicolor* genotypes for all the four antioxidant metrics evaluated in this work (Fig 2). As wild sorghums generally show a higher antioxidant content compared to domesticated ones [20], *S. halepense* alleles can contribute significantly to this category of traits, especially those, such as tannins, which have been selected against during domestication.

In our study we found significant differences between allele frequencies when the two sub-groups SB and SBxSH were compared. On average, the alternative allele frequency was higher in SBxSH than in SB (Fig 6). This evidence might to some extent reflect the narrowing of the genetic base and the consequent reduction of variability associated with sorghum domestica-tion; however, two considerations must be taken into account. First, allele frequencies are

computed considering a 1:1 ref:alt allele ratio at heterozygous loci, as is the case for diploid genotypes; however, *S. bicolor × S. halepense* hybrid lines are expected to be tetraploid. Therefore, possible allele ratios of 3:1 and 1:3 can introduce a bias in such calculation for heterozygous genotypes. Unfortunately, these alleles ratios could not be evaluated due to the low coverage, as it would require a number of reads/locus high enough to support a reliable determination of allele dosage; this latter aspect is however considered out of the scope of this work. The second relevant aspect to consider is the alignment of reads from tetraploid individuals to the reference genome sequence of a diploid inbred line. This implies the alignment of homeologs to the same locus, which results in an overestimation of heterozygosity. Homeologs in *S. halepense* are expected to descend from orthologs in the genomes of *S. bicolor* and *S. propinquum*, being these two species considered its ancestors. The fate of these homeologs in *S. bicolor x S. halepense* hybrid lines after several generations is very difficult to predict, given the different possibilities of chromosome pairing at meiosis [70]; however, some of the homeolog chromosome pairs can be maintained and contribute to increasing the genetic variability of hybrid lines.

The panel of *S. bicolor × S. halepense* hybrid lines analyzed in this study is indeed small, yet some significant associations were detected. Among the 57 markers that were found to be associated to one or more of the analyzed traits, the majority (72%) were polymorphic in both SB and SBxSH subgroups (Table 2), although allele frequencies were in several cases different, consistently with what observed on the entire dataset of markers (Fig 6). 13 significant markers were polymorphic only in SB, most likely due to the small number of *S. bicolor* lines used as parents in SBxSH hybridization, which could retain only a portion of the genetic variability available in domesticated sorghum. Finally, 3 significant markers (Chr1_20707841, Chr2_13905455 and Chr3_57670375) were polymorphic only in SBxSH lines, highlighting the contribution of the *S. halepense* genome to GWAS in spite of the small size of the SBxSH group. Not surprisingly, two of these markers (on chromosomes 2 and 3) resulted significantly associated to tannins, whose content is expected to be lower in domesticated sorghum due to human selection traditionally oriented against this trait. Noteworthy, the marker Chr2_13905455 was among the major effect loci ($R^2$ 20.75) and registered the highest positive effect of the entire dataset (+5710, Table 2). Its position is consistent with other studies reporting this region on chromosome 2 as associated with grain color [31,32], and the candidate gene Sobic.002G117500.1 (similar to an UDP-flavonoid glucosyl transferase and to A. thaliana TT15), placed nearly 600Kbp far from marker Chr2_13905455, was proposed to underlie its effect on phenotype [31]. In our study, however, a closer candidate, Sobic.002G115700.1, was found at only 330 Kbp from the marker; this gene shows homology to a chalcone synthase and *A. thaliana* TT4 (S1 Table) involved in the biosynthesis of polyphenols [31,32]. Clearly our results suggest that *S. halepense* provided a new allele at this locus with a remarkably high effect on condensed tannins (proanthocyanidins) in sorghum grain. The SBxSH lines evaluated in this work can therefore be considered as a good source of antioxidants variability in addition to the possibility of using them to breed for perennial grain sorghum fortified with condensed tannins. The introgression of *S. halepense* into *S. bicolor* to develop novel varieties was effectively accomplished in biomass sorghum breeding programs [8]. Nonetheless, the use of *S. halepense* for improving sorghum grain nutritional quality is expected to be challenging particularly due to the necessity to eliminate negative wild-related traits such as loose panicle, tiny seeds, and seed chattering. Moreover, it is reasonable to hypothesize that using a larger SBxSH population could lead to the identification of new alleles and even new, additional loci associated to these traits.

The purpose of this work was to conduct GWAS of sorghum grain antioxidants using SNPs in a novel diversity panel of *Sorghum bicolor* landraces and *S. bicolor × S. halepense*

recombinant inbred lines. To the best of our knowledge, it is the first time such a panel is used in association analysis. *Sorghum halepense* contributed novel polymorphism, and sorghum recombinant inbred lines derived from *Sorghum bicolor* x *Sorghum halepense* controlled hybridizations outperformed *Sorghum bicolor* landraces for antioxidant production. This highlights the importance of *S. halepense* genome not only as a source of perenniality but also as a donor of genetic factors for antioxidant production. Antioxidant traits were perfectly highly correlated and showed very high broad-sense heritability. The genome-wide association analysis conducted in this work uncovered several major effect and novel QTLs explaining higher proportion of the variability that existed in the antioxidant traits than in previous works. These QTLs can be directly used in marker assisted breeding and/or validated using different approaches including linkage mapping before tools like breeder's chip can be produced for large-scale uses in breeding programs. The GWAS results presented herein and experimental designs used in this work can be implemented in antioxidants genetic investigations and in breeding programs to qualitatively and quantitatively improve the antioxidant production for different purposes including the manufacture of health-promoting and specialty foods.

## Supporting information

**S1 Table. Annotated genes harboring major effect markers ($R^2 \geq 15\%$).** Highlighted in green are genes annotated from Rhodes et al. 2014,2017, in orange genes annotated as similar to Peroxidase, in yellow new annotations from sorghum genome in Atlas. In the first three columns start and stop position on the sorghum genome and transcript name, followed by the nearest marker name and the distance of the gene from the nearest marker, then a column where are shown the GWAS methods and target traits for which the linked SNP was significant, the last column shows the category of the genes.
(DOCX)

**S2 Table. 'A posteriori' genes harboring major effect marker ($R^2 \geq 15\%$).**
(DOCX)

## Acknowledgments

The authors would like to thank the reviewers and editor for their informative remarks that contributed to improving the quality of this paper.

## Author Contributions

**Conceptualization:** Ephrem Habyarimana.

**Data curation:** Ephrem Habyarimana, Michela Dall'Agata, Paolo De Franceschi.

**Formal analysis:** Ephrem Habyarimana.

**Funding acquisition:** Ephrem Habyarimana.

**Investigation:** Ephrem Habyarimana.

**Methodology:** Ephrem Habyarimana, Michela Dall'Agata, Paolo De Franceschi, Faheem S. Baloch.

**Project administration:** Ephrem Habyarimana.

**Software:** Ephrem Habyarimana.

**Supervision:** Ephrem Habyarimana.

**Writing – original draft:** Ephrem Habyarimana.

**Writing – review & editing:** Ephrem Habyarimana, Michela Dall'Agata, Paolo De Franceschi, Faheem S. Baloch.

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
