## [Decision Letter · Decision Letter 0]

22 Oct 2019

PONE-D-19-23594

Genome-wide association mapping of total antioxidant capacity, phenols, tannins, and flavonoids in a panel of Sorghum bicolor and S. bicolor × S. halepense populations using multi-locus models

PLOS ONE

Dear Dr. Habyarimana,

Thank you for submitting your manuscript to PLOS ONE. After careful consideration, we feel that it has merit but does not fully meet PLOS ONE’s publication criteria as it currently stands. Therefore, we invite you to submit a revised version of the manuscript that addresses the points raised during the review process.

the reviewer has gone through the manuscript and have very constructive comments which can help improve the manuscript. I encourage the authors to go through these comments in detail and address as much as possible.

We would appreciate receiving your revised manuscript by Dec 06 2019 11:59PM. To enhance the reproducibility of your results, we recommend that if applicable you deposit your laboratory protocols in protocols.io, where a protocol can be assigned its own identifier (DOI) such that it can be cited independently in the future. For instructions see: http://journals.plos.org/plosone/s/submission-guidelines#loc-laboratory-protocols

We look forward to receiving your revised manuscript.

Kind regards,

Ajay Kumar

Academic Editor

PLOS ONE

Journal Requirements:

https://www.sciencedirect.com/science/article/pii/S0961953417300910?via%3Dihub

https://onlinelibrary.wiley.com/doi/abs/10.1094/CCHEM-03-16-0075-R

http://www.ogtr.gov.au/internet/ogtr/publishing.nsf/Content/5DCF28AD2F3779C4CA257D4E001819B9/$File/Sorghum%20Biology%20Version%201.1%20July%202017.pdf

https://pubs.acs.org/doi/abs/10.1021/jf503651t

In your revision ensure you cite all your sources (including your own works), and quote or rephrase any duplicated text outside the methods section. Further consideration is dependent on these concerns being addressed.

5. Please include a caption for figure 7.

Additional Editor Comments (if provided):

Reviewers' comments:

Reviewer's Responses to Questions

**Comments to the Author**

1. Is the manuscript technically sound, and do the data support the conclusions?

Reviewer #1: Partly

2. Has the statistical analysis been performed appropriately and rigorously? 

Reviewer #1: I Don't Know

3. Have the authors made all data underlying the findings in their manuscript fully available?

Reviewer #1: Yes

4. Is the manuscript presented in an intelligible fashion and written in standard English?

Reviewer #1: Yes

5. Review Comments to the Author

Reviewer #1: Manuscript needs structure: analysis need details and designated headings. Analysis for population structure and GWAS are explained under Results for some reason.

In general, Introduction didn't have good structure and flow.

Materials and Methods lacked references, and details on statistical analysis.

Discussion includes statistical analysis. Stats should belong to Materials and Methods.

Results are very general. Results are missing phenotypic data analysis.

Conclusion is absent.

I went line by line with small comments but please go over the whole manuscript to change similar comments addressed. More references need to be added to statements, many sentences start with abbreviations and/or numbers. Make sure all abbreviations are explained before they are used.

Measure units are not consistently separated with numbers……sometimes there is a space between, sometimes there is not. Numbers from one to ten are spelled out (line 492).

ABSTRACT

Abstract doesn’t state the importance of the study. Abstract has to be re-written in an easier to digest more informative form with no sentences starting with abbreviations and numbers (see below for details). All abbreviations have to be spelled out prior to using them. No need to use name of the QTLs in the abstract (created clutter).

Line 19: use full form of GWAS, SNPs, before their abbreviation.

Line 21: Don’t start the sentence with abbreviation (RILs) identify R2 - what R2 in "A novel pleiotropic major effect marker Chr1_61095994 showed highest (27-31%) R2 27 in most traits, and was in LD with a hotspot of 19 putative GST genes 28 conjugating anthocyanins into vacuoles."

Line 22: what does it mean “Antioxidant traits were perfectly correlated” ?

Line 23: Don’t start new sentence with abbreviation (GWAS)

Line 24: Abbreviation QTLs was not spelled out earlier in the text

Line 25: don’t start new sentence with the number

Line 25-26: this sentence is not informative

Line 27: R2 has to be explained

Line 27: GST?

Line 26-30: Sentences out of content…. flavonoids were not mentioned before. Don’t start sentence with “another”.

Line 29: MBW?

Line 30: Results are very generalized

Introduction

Line 37-41: Sentences need to be broken down to a smaller ones. At the moment different statements are put together and separated with a comma.

Line 41-43: No connection from previous sentences. Need more genetic story on Sorghum before started on the genome size. Also need reference.

Start on different types of sorghum: perennial vs annual, etc. Compare their genetics?

43: Reference needed

45-55: what is the importance of the information?

64-65: Review pericarp structure. Testa is the part of a pericarp layer.

68: References

76: References

83: Reference for the following statement needed “Given that one of the objectives of breeding has long been to select against the tannins in the caryopsis, particularly because of its astringency and its negative effect in animal feeding, the persistent residual tannin trait n cultivated sorghum needs to be explained.”

89-90: Reference for “Conventionally, grain color was used as a proxy to indirectly select against or for polyphenols in sorghum.”

92-97: References

102-104: It’s an assumption, remove “doubtless”. Add reference.

108 – References

Materials and Methods:

145-151: When referencing methods please use original source of the methods and not the papers where the original source were mentioned

150: Don’t start a sentence with abbreviation SB

Materials and Methods: how many replications? 2 years, one location.

164: No AACCI methods were used for Total phenols, tannins, flavonoids and antioxidant capacity (TAC) determination? If they were, please reference to them.

166 : Add manufacturer

171: Don’t start with abbreviation

173: Statistical analysis part should be under separate heading. Not enough information on what statistical analysis were performed and how. For example, what P F-test was significant?

My concern is I didn’t catch if phenol, TAC were measured as is or per kernel? It would make a tremendous difference if the seeds are smaller in some of the lines, it takes more of them to make a gram, therefore TAC capacity would be greater as is but would be smaller per kernel.

Again, statistical analysis need to be addressed more fully/detailed and preferably under separate heading.

192: Abbreviation GBS in the heading. Was not mentioned earlier.

207: Put comma instead on dot in the number of SNP used

267: Don’t start sentence with abbreviation

All the best with corrections and publication!!

6. PLOS authors have the option to publish the peer review history of their article (what does this mean?). If published, this will include your full peer review and any attached files.

Reviewer #1: Yes: Marina Johnson

---

## [Author Response · Author response to Decision Letter 0]

29 Oct 2019

Rebuttal letter that responds to each point raised by the academic editor and reviewer

Dear Editor, 

It was great for my Lab to be able to submit our findings to PLOS ONE Journal for which we recognize the high standards. 

We are heartfully thankful to Editor and Reviewer for informative and constructive review and suggestions to improve our manuscript. 

We have gone through all the comments of the reviewer and please find below our responses as highlighted in light blue font color. 

Reviewer #1: Manuscript needs structure: analysis need details and designated headings. Analysis for population structure and GWAS are explained under Results for some reason.

Response: we reviewed the manuscript and improved its structure. We have added a designated heading for the statistical analyses. We found it useful to explain why we chose to present (under the result section) the genomic relationship matrix instead of principal component analysis. GWAS approaches implemented in this work was amply (Line 228 - 281) described under the Materials and Methods section. 

In general, Introduction didn't have good structure and flow. 

Response: we greatly improved the flow of information throughout the manuscript. 

Materials and Methods lacked references, and details on statistical analysis. 

Response: we added reference as necessary as we could, and we added a heading for statistical analyses

Discussion includes statistical analysis. Stats should belong to Materials and Methods. 

Response: Statistical analyses were described under materials and methods, and the results were presented under the results section. Under the discussion section we discussed our results in order to put the reader into the context of our findings. From time to me as we judged useful to highlight key findings, we included corresponding value, for instance. 

Results are very general. Results are missing phenotypic data analysis. 

Response: We respectfully disagree with the reviewer: The results were presented to specifically address the objectives of the study; we presented our results in concise, sequential, and integrated way. Phenotypic analyses were presented in the form of quantile analysis using boxplot and we also computed the Tukey honestly significant difference test (refer to Fig. 2, for instance). 

Conclusion is absent. 

Response: the section “Conclusion” is optional in PLOS ONE according to current guidelines (), and indeed we judged it was not necessary. But we included a concluding paragraph starting from Line 761 and ending at Line 777. 

I went line by line with small comments but please go over the whole manuscript to change similar comments addressed. More references need to be added to statements, many sentences start with abbreviations and/or numbers. Make sure all abbreviations are explained before they are used. 

Response: we went through the manuscript and made corrections

Measure units are not consistently separated with numbers……sometimes there is a space between, sometimes there is not. Numbers from one to ten are spelled out (line 492). 

Response: we went through the manuscript and made corrections

ABSTRACT

Abstract doesn’t state the importance of the study. Abstract has to be re-written in an easier to digest more informative form with no sentences starting with abbreviations and numbers (see below for details). All abbreviations have to be spelled out prior to using them. No need to use name of the QTLs in the abstract (created clutter).

Response: we extensively edited the abstract to include importance of the study, making it easier to digest. Abbreviation issue was solved as suggested by the reviewer. We nonetheless judged informative to maintain the names of the few outstanding QTLs within the abstract without compromising the conciseness, clarity and simplicity. 

Line 19: use full form of GWAS, SNPs, before their abbreviation.

Response: corrected

Line 21: Don’t start the sentence with abbreviation (RILs) identify R2 - what R2 in "A novel pleiotropic major effect marker Chr1_61095994 showed highest (27-31%) R2 27 in most traits, and was in LD with a hotspot of 19 putative GST genes 28 conjugating anthocyanins into vacuoles."

Response: we corrected and spelled out the R2.

Line 22: what does it mean “Antioxidant traits were perfectly correlated” ?

Response: we edited to “highly”. 

Line 23: Don’t start new sentence with abbreviation (GWAS)

Response: we made correction.

Line 24: Abbreviation QTLs was not spelled out earlier in the text

Response: we spelled out QTLs

Line 25: don’t start new sentence with the number

Response: we corrected

Line 25-26: this sentence is not informative

Response: we respectfully disagree with the reviewer; the sentence is meaningful as we are stating quantitatively about our findings.

Line 27: R2 has to be explained

Response: we explained

Line 27: GST?

Response: GST was spelled out

Line 26-30: Sentences out of content…. flavonoids were not mentioned before. Don’t start sentence with “another”.

Response: flavonoids were explained and “another” was removed

Line 29: MBW?

Response: MBW was spelled out as it is presented in scientific format.

Line 30: Results are very generalized

Response: we respectfully disagree with the reviewer. This is abstract and we highlighted the importance of the work carried out, and we sequentially presented our findings and corresponding interests. We added “for the development of sorghum cultivars with consumer-tailored antioxidants concentration” to clearly highlight our vision. 

Introduction

Line 37-41: Sentences need to be broken down to a smaller ones. At the moment different statements are put together and separated with a comma.

Response: we respectfully disagree with the reviewer. The sentence is not too long and we added commas to make it easily understood.

Line 41-43: No connection from previous sentences. Need more genetic story on Sorghum before started on the genome size. Also need reference.

Response: we agree with the reviewer, and we removed the sentence.

Start on different types of sorghum: perennial vs annual, etc. Compare their genetics?

Response: the flow of ideas was improved by removed the unnecessary sentence above.

43: Reference needed

Response: the reference was no longer needed as the above sentence was removed. 

45-55: what is the importance of the information?

Response: we made our statements clearer, which made the importance straightforward. We showed the importance of using Sorghum halepense in controlled hybridizations with Sorghum bicolor, and made a smooth transition to the ongoing endeavor in our breeding program, and then transitioned to the importance of antioxidants that can also be improved by the above mentioned hybridizations. 

64-65: Review pericarp structure. Testa is the part of a pericarp layer. 

Response: we respectfully disagree with the reviewer. Pericarp is the outermost structural component of the caryopsis and is composed of three sublayers, namely epicarp, mesocarp and endocarp. Just underneath the endocarp is the testa layer or seed-coat (http://www.fao.org/3/t0818e/t0818e02.htm)

68: References

Response: added

76: References 

Response: added

83: Reference for the following statement needed “Given that one of the objectives of breeding has long been to select against the tannins in the caryopsis, particularly because of its astringency and its negative effect in animal feeding, the persistent residual tannin trait n cultivated sorghum needs to be explained.”

Response: added

89-90: Reference for “Conventionally, grain color was used as a proxy to indirectly select against or for polyphenols in sorghum.”

Response: added

92-97: References

Response: added

102-104: It’s an assumption, remove “doubtless”. Add reference.

Response: removed “doubtless” and added reference

108 – References

Response: added [33]

Materials and Methods:

145-151: When referencing methods please use original source of the methods and not the papers where the original source were mentioned 

Response: reference [11] added

150: Don’t start a sentence with abbreviation SB

Response: we added “Sorghum bicolor”

Materials and Methods: how many replications? 2 years, one location. 

Response: Yes it is 2 years, one location (but different sites as dictated by the necessity of crop rotation), and we added: “with 6 controls (checks) and 6 blocks”

164: No AACCI methods were used for Total phenols, tannins, flavonoids and antioxidant capacity (TAC) determination? If they were, please reference to them. 

Response: No

166 : Add manufacturer

Response: added “UDY Corporation”

171: Don’t start with abbreviation

Response: corrected and reviewed throughout the text

173: Statistical analysis part should be under separate heading. Not enough information on what statistical analysis were performed and how. For example, what P F-test was significant? 

Response: a separate heading for antioxidants statistical analysis was created and called: “Antioxidants Statistical Analysis”. As for other minor details such as significance level, they were included under in the appropriate captions. 

My concern is I didn’t catch if phenol, TAC were measured as is or per kernel? It would make a tremendous difference if the seeds are smaller in some of the lines, it takes more of them to make a gram, therefore TAC capacity would be greater as is but would be smaller per kernel. 

Response: TAC was expressed as mmol TE (Trolox equivalents) kg-1 dry mass basis whereas phenols were expressed as gallic acid equivalents (g GAE kg-1 dry mass basis)

Again, statistical analysis need to be addressed more fully/detailed and preferably under separate heading. 

Response: done as described above

192: Abbreviation GBS in the heading. Was not mentioned earlier. 

Response: GBS was spelled out

207: Put comma instead on dot in the number of SNP used

Response: comma added

267: Don’t start sentence with abbreviation

Response: we added “The statistical inferences showed that” before the abbreviation

All the best with corrections and publication!!

Response: We heartfully thank the reviewer and editor for constructive and informative review and suggestions.

---

## [Decision Letter · Decision Letter 1]

18 Nov 2019

Genome-wide association mapping of total antioxidant capacity, phenols, tannins, and flavonoids in a panel of Sorghum bicolor and S. bicolor × S. halepense populations using multi-locus models

PONE-D-19-23594R1

Dear Dr. Habyarimana,

We are pleased to inform you that your manuscript has been judged scientifically suitable for publication and will be formally accepted for publication once it complies with all outstanding technical requirements.

With kind regards,

Ajay Kumar

Academic Editor

PLOS ONE

Additional Editor Comments (optional):

Reviewers' comments:

Reviewer's Responses to Questions

**Comments to the Author**

1. If the authors have adequately addressed your comments raised in a previous round of review and you feel that this manuscript is now acceptable for publication, you may indicate that here to bypass the “Comments to the Author” section, enter your conflict of interest statement in the “Confidential to Editor” section, and submit your "Accept" recommendation.

Reviewer #1: All comments have been addressed

2. Is the manuscript technically sound, and do the data support the conclusions?

Reviewer #1: Yes

3. Has the statistical analysis been performed appropriately and rigorously? 

Reviewer #1: Yes

4. Have the authors made all data underlying the findings in their manuscript fully available?

Reviewer #1: Yes

5. Is the manuscript presented in an intelligible fashion and written in standard English?

Reviewer #1: Yes

6. Review Comments to the Author

Reviewer #1: Authors addressed comments and has improved the manuscript. I would run it through a proof reader professional anyway to ensure minor grammar, punctuation and typos are eliminated however.

7. PLOS authors have the option to publish the peer review history of their article (what does this mean?). If published, this will include your full peer review and any attached files.

Reviewer #1: Yes: Marina Johnson

---

## [Editor Report · Acceptance letter]

22 Nov 2019

PONE-D-19-23594R1 

Genome-wide association mapping of total antioxidant capacity, phenols, tannins, and flavonoids in a panel of *Sorghum bicolor *and *S. bicolor *× *S. halepense *populations using multi-locus models 

Dear Dr. Habyarimana:

I am pleased to inform you that your manuscript has been deemed suitable for publication in PLOS ONE. Congratulations! Your manuscript is now with our production department. 

With kind regards,

on behalf of

Dr. Ajay Kumar 

Academic Editor

PLOS ONE